# Biomass Burning CO Emissions: Exploring Insights through TROPOMI-derived Emissions and Emission Coefficients

Debora Griffin[1], Jack Chen[1], Kerry Anderson[1,2], Paul Makar[1], Chris A. McLinden[1,3], Enrico Dammers[4], and Andre Fogal[1,5]

[1]Air Quality Research Division, Environment and Climate Change Canada, Toronto, Ontario, Canada
[2]Natural Resources Canada (emeritus)
[3]Department of Physics and Engineering Physics, University of Saskatchewan, Saskatoon, Saskatchewan, Canada
[4]Netherlands Organisation for Applied Scientific Research (TNO), Climate Air and Sustainability (CAS), Utrecht, The Netherlands
[5]University of Waterloo, Department of Physics and Astronomy, Waterloo, Ontario, Canada

**Correspondence:** D. Griffin (debora.griffin@ec.gc.ca)

**Abstract.** Emissions from biomass burning are a significant source of air pollution, which can adversely impact air quality and ecosystems thousands of kilometers downwind. These emissions can be estimated by a bottom-up approach that relies on fuel consumed and standardized emission factors. Emissions are also commonly derived with a top-down approach, using satellite observed fire radiative power (FRP) as proxy for fuel consumption. biomass burning emissions can also be estimated directly from satellite trace gas observations, including carbon monoxide (CO). Here, we explore the potential of satellite-derived CO emission rates from biomass burning and provide new insights into the understanding of satellite-derived fire CO emissions globally, with respect to differences in regions and vegetation type. Specifically, we use the TROPOMI (Tropospheric Monitoring Instrument) high spatial-resolution satellite datasets to derive burning CO emissions directly for individual fires between 2019 and 2021 globally. Using synthetic data (with known emissions) we show that the direct emission estimate methodology has a 34 % uncertainty for deriving CO emissions (and a total uncertainty of 44 % including wind and CO column uncertainty). From the TROPOMI-derived CO emissions we derive biome specific emission coefficients (emissions relative to FRP) by combining the direct emission estimates and the satellite observed FRP from the Moderate Resolution Imaging Spectrometer (MODIS). These emission coefficients are used to establish annual top-down CO emission inventories from biomass burning showing that southern hemisphere Africa has the highest CO biomass burning emissions (over 25 % of global total of 300-390 Mt(CO)/yr between 2003-2021), and almost 25 % of global CO biomass burning emissions are from broad-leaved evergreen tree fires. A comprehensive comparison between direct estimates, top-down and bottom-up approaches, provides insight into the strengths and weaknesses of each method: FINN2.5, has higher CO emissions, by a factor between two and five, compared to all other inventories assessed in this study. Trends over the past two decades are examined for different regions around the globe showing that global CO biomass burning emissions have, on the whole, decreased (by 5.1 to 8.7 Mt(CO)/yr), where some regions experience increased and others decreased emissions.

# 1 Introduction

Emissions from biomass burning are a significant source of air pollution in the global atmosphere. These emissions are transported over large distances and, thus, can adversely impact air quality and ecosystems thousands of kilometers downwind (e.g., Landis et al., 2018; Meng et al., 2019). Health impacts are typically more severe in close proximity to the fires, however health impacts from transported smoke plumes have also been reported (Matz et al., 2020). In more recent years an increase in fire activity in North America has been recorded (e.g., Romero-Lankao et al., 2014; Landis et al., 2018). The driving factors for this increase (in North America) include droughts, higher temperatures, and fuel loading caused by tree death (Littell et al., 2009; Westerling, 2016). This trend may continue due to climate change (Liu et al., 2013; Wotton et al., 2017). Biomass burning emissions are associated with large uncertainties (Andreae, 2019), which lead to a growing demand for improved knowledge of biomass burning emissions.

Biomass burning emissions, e.g. from wildfires can be estimated by either a bottom-up or top-down approach. For the bottom-up estimates proxies, such as fuel type, area burned, and emission factors (EF), are used to determine the emissions; these emissions are determined by

$$E_i = \text{Activity} \times EF_{i,f}. \tag{1}$$

Where the emissions ($E_i$) account for mass of fuel consumed in combustion (kg), which are based on fire activity (which includes factors such as the burned area, fuel loading, fuel classification, and a combustion factor), and $EF_{i,f}$ (g/kg) is the emission factor for a specific chemical species ($i$), which is typically a function of fuel type ($f$) (sometimes it can be dependent on the combustion type as well, i.e., flaming and smoldering). An example for this type of fire emission inventory is the Fire INventory from NCAR (FINN; Wiedinmyer et al., 2011). Biomass Burning emissions are also commonly derived with a top-down approach, using satellite information to estimate biomass burning emissions, sometimes in near-real time. The Global Fire Assimilation System (GFAS; Kaiser et al., 2012)) is an example of such a system, wherein satellite observed fire radiative power (FRP, in units of W) or its time integral, fire radiative energy (FRE, in units of J) is used as as proxy for fuel consumption. Using a satellite-remote sensing FRP together with a species specific emissions coefficient (EC$_i$) is a common approach for top-down fire inventories: $E_i = \text{FRE} \times \text{EC}_i$. For many inventories, $\text{EC}_i$ (in units of g/MJ) is commonly estimated from $\text{EF}_{i,f}$ (dependant on species and fuel type, in units of g/kg) and a conversion factor $\beta_f$ (kg/MJ) based on fuel type (Kaiser et al., 2012). The $\text{EF}_{i,j}$ is typically based on laboratory derived values and is a common factor used in both top-down and bottom-up approaches. Other inventories make use of a combination of bottom-up and top-down information (e.g. Global Fire Emissions Database (GFED; Giglio et al. (2013)). The Canadian Forest Fire Emissions Prediction System (CFFEPS; Chen et al. (2019)) and its global extension Global Forest Fire Prediction System (GFFEPS; under-development, (Anderson et al., 2024)), where satellite-derived hotspot data is linked to databases such as fuel type, previous statistics on area burned per hotspot for a given fuel type, etc., to determine emissions, and may be run within an on-line air-quality model to determine the effects of fire emissions on weather (Makar et al., 2020).

Fire emission rates can also be derived directly from measurements without the proxy information of combustion processes. This is an alternative measure for evaluation of emission inventories and emission processing systems. In situ and aircraft measurements are difficult to obtain close to fires and may be rare due to the expense of observations. However, satellite-borne observations provide ongoing coverage using a common instrument platform, which can thus be used to constrain biomass burning emissions. Satellite-remote sensing observations also have the advantage of near-global coverage. With the recent

advances in satellite observations, biomass burning emissions can be estimated directly, with past work showing the utility of these observations in estimating emissions of carbon monoxide (CO) (Adams et al., 2019; Stockwell et al., 2022), carbon dioxide (CO2) (Guo et al., 2019), nitrogen oxides ($NO_x$) (Jin et al., 2021; Griffin et al., 2021), and ammonia ($NH_3$) (Adams et al., 2019). There are several limitations to emission estimates from satellite observations: 1) direct satellite-based emissions estimates are only possible for a few chemical species that are measured by satellite instruments; 2) current satellite-based

instruments are on polar-orbiting platforms that consequently only observe a location once or twice per day, a measurement frequency that will improve in the near future with new geostationary satellites; 3) many fires are missed due to cloud cover or even thick smoke that impacts the quality of the satellite observation; 4) small fires that are below the satellite detection limit will be missed. Thus, direct estimates are a great tool to derive emissions at the time of the overpass for specific fires, but cannot be used alone to determine a total emission inventory (i.e., only one point per day, many missed fires).

In this study CO biomass burning emissions are directly derived from the satellite observations using the TROPOMI (Tropospheric Monitoring Instrument) high spatial-resolution satellite observations on CO and plume height information, and the uncertainty of this method is assessed using synthetic CO columns (with known emissions). The main advantage of the TROPOMI dataset is the wealth of observations at higher horizontal resolution (with $7 \times 5.5\,\mathrm{km}^2$, $7 \times 7\,\mathrm{km}^2$ before August

2019) compared to its predecessors (e.g., the Infrared Atmospheric Sounding Interferometer (IASI; Clerbaux et al. (2009)), and Measurement of Pollution in the Troposphere (MOPITT; Deeter et al. (2013))). Another advantage of TROPOMI over its predecessors is the high sensitivity near the surface (Schneising et al., 2020) which is beneficial when estimating emissions that occur close to the surface. TROPOMI has previously been used to estimate CO emissions from wildfires and comparisons to aircraft-derived emissions showed very good agreement for fires in North America (Stockwell et al., 2022). It has also been

used to derive fire emissions in Portugal (Magro et al., 2021). Most recently, Goudar et al. (2023) published an automated plume detection and emission estimation algorithm utilizing TROPOMI CO, in our study, an alternative approach is explored.

The aim of this study is to establish a database of direct satellite-derived biomass burning CO emissions globally; this approach has been entirely automated and has the capability to determine CO fire emissions in quasi near real time (as soon as

TROPOMI CO and MODIS FRP observations are available). Additionally, we determine biome specific emission coefficients (ECs; emissions relative to FRP), which are based on the direct emissions from TROPOMI relative to the amount of heat energy released by the fire (FRP) from MODIS, and ultimately establish top-down annual total emissions based on these derived ECs. These emission coefficients can provide insights into the efficiency of combustion, and help quantifying how emissions from a particular ecological region or biome are related to the heat energy generated by biomass burning in that region. This

information can be valuable for understanding the environmental impact of biomass burning in different ecosystems and for developing strategies to manage and mitigate their effects. Furthermore, FRP is often and more easily measured from satellites compared to CO, and determining a biome specific CO-to-FRP ratio can help to determine the daily total emissions of fires. We further show how combining the satellite derived CO emissions with satellite observed FRP from the Moderate Resolution Imaging Spectrometer (MODIS) can establish an annual CO emission inventory from biomass burning, by applying the derived emission coefficients to assimilated daily FRP based on MODIS measurements (available from GFAS). As we show below, the direct emissions as well as the alternative annual inventory (from the here derived ECs) provides an alternative measure that can be used to evaluate and improve fire emission inventories or fire emissions prediction systems.

This paper is structured as follows: Section 2 describes the datasets and the direct emission estimation algorithm used, and includes an evaluation of the method with synthetic columns. The direct TROPOMI-derived emissions are compared to GFFEPS bottom-up emissions in Sect. 3. In Sect. 4, the emissions coefficients and burning efficiency from different biomes are discussed. Annual global inventories of CO fire emissions from different inventories are compared and evaluated against the satellite-derived CO emissions estimates (using ECs), including a trend analysis over the past two decades in Sect. 5, followed by a summary and conclusions in Sect. 6.

## 2    Datasets and Methods

### 2.1    Satellite CO Dataset

The TROPOMI instrument, on-board the Copernicus Sentinel-5 Precursor (S-5P) satellite (under ESA), orbits the globe with a local overpass time of around 13:30 local time and near full-surface coverage on a daily basis (Veefkind et al., 2012; Hu et al., 2018). It has four spectrometers that cover the solar spectrum between the short-wave infra-red (SWIR) and the ultra-violet (UV). Amongst other species, total column CO is retrieved from the SWIR spectrometer at a horizontal resolution of roughly $5.5 \times 7\,\text{km}^2$ ($7 \times 7\,\text{km}^2$ prior to August 6, 2019) using the Shortwave Infrared CO Retrieval algorithm (Borsdorff et al., 2018, 2019). The TROPOMI CO columns have been validated with satellite observations (Martínez-Alonso et al., 2020), as well as with ground-based remote sensing instruments (Borsdorff et al., 2019). Both studies showed that TROPOMI exceeded its mission requirement on precision and concluded a precision error well below <10 %. Validation against TCCON measurements around the world showed that the TROPOMI CO columns have a high bias of about 10 % (Sha et al., 2021). For our analysis, we have utilized observations rated with a quality flag greater than 0.5, where 0 represents the lowest quality and 1 denotes the highest quality. This choice aligns with the recommended quality threshold (Apituley et al., 2018). Notably, when we investigate areas near active fires, the quality flag of the retrieval can be impacted by the presence of smoke. Consequently, only including observations with a quality flag of 1 would result in the exclusion of a substantial number of data points, primarily due to the influence of smoke (we found most pixels inside smoke plumes have a quality flag value of 0.7).

The CO averaging kernel from the TROPOMI observations predominantly registers values close to 1 within the boundary layer for cloud-free conditions, specifically around 0.95 with a narrow range of variability (approximately ±0.05) (Schneising et al.,

2020). Nevertheless, the presence of clouds diminishes the sensitivity of the averaging kernel beneath them. It is important to note that smoke is primarily comprised of fine minuscule particles ($\sim$0.25 $\mu m$ or smaller (Junghenn Noyes et al., 2022)).
At a wavelength of 2.3$\mu$m these particles scatter minimal light. Looking at the TROPOMI averaging kernel, we found that in case of fires the sensitivity close to the surface is typically lower than 1. Rowe et al. (2022) investigated TROPOMI CO in thick fire plumes and found agreement within 13-16 % (Table 3) without considering the averaging kernel which has been used to estimate the overall uncertainty of the emissions. The effect of the averaging kernel depends on 1) the shape of the averaging kernel and 2) on the CO profile, looking at different profiles and averaging kernels, we found the largest effect is for an averaging kernel that is close to 0 at the surface and for a strong enhancement of the CO profile, the magnitude of this enhancement determines the magnitude of the averaging kernel corrected columns. We found that the effect that applying the averaging kernel inside the smoke plume always increases the CO columns. Other than Rowe et al. (2022) (who investigated fires during FIREX-AQ) we do not have profiles (globally) that can be applied for an averaging kernel correction. Testing the effect on the emissions we used GEM-MACH model profiles and applied the averaging kernel correction (see Appendix Fig. B1) which showed a 17 % increase of the emissions for one specific fire.

## 2.2 Satellite-derived CO Emissions

To determine emissions from the satellite observations, a similar approach based on Adams et al. (2019) and as described in Stockwell et al. (2022) and Griffin et al. (2021) was applied. The basic underlying concept of the method is mass balance: that the source rate Q must be equivalent to the product of the column plume transect and the wind speed (U). The mass within the box can be determined by integrating the enhanced vertical column densities (VCDs) over the background concentrations and applying the molar mass of CO. The time of a mass clearing the box is based on the length of the box and the wind speed. This methodology has been used previously to determine emissions from biomass burning plumes using satellite observations (e.g. Mebust et al., 2011; Mebust and Cohen, 2014; Adams et al., 2019; Griffin et al., 2021; Stockwell et al., 2022).

The flux methodology employed here is best applied to emitted species with slow chemical loss rates such as CO, because the flux method is insensitive to the plume shape (often the plumes are not flawless Gaussian distributions, especially for long-lived gases). Further, specifically for estimating fire emissions at a time when the fire activity is increasing (i.e. TROPOMI overpass time at 1:30 pm) will impact any attempt to estimate the chemical life-time for shorter-lived species like $NO_2$ (Griffin et al., 2021) from the flux method, this however will not impact the analysis for a long-lived species like CO, as the lifetime is known. The removal of the background CO is, however, very important in the flux method, and may otherwise influence the estimated emission rate significantly, as the CO background is relatively high.

The approach described as the fitting method, is summarized here. New improvements with regards to the plume rotation and plume widths are included, and illustrated in Fig. 1 where Gaussians are fitted across the plume to be able to automate the estimation by determining the plume width, correct the wind direction and correct the centre location of the fire. First, a binned upwind/downwind domain of a regular grid size (4 km x 4km) is established by a wind-rotation about the approximate center

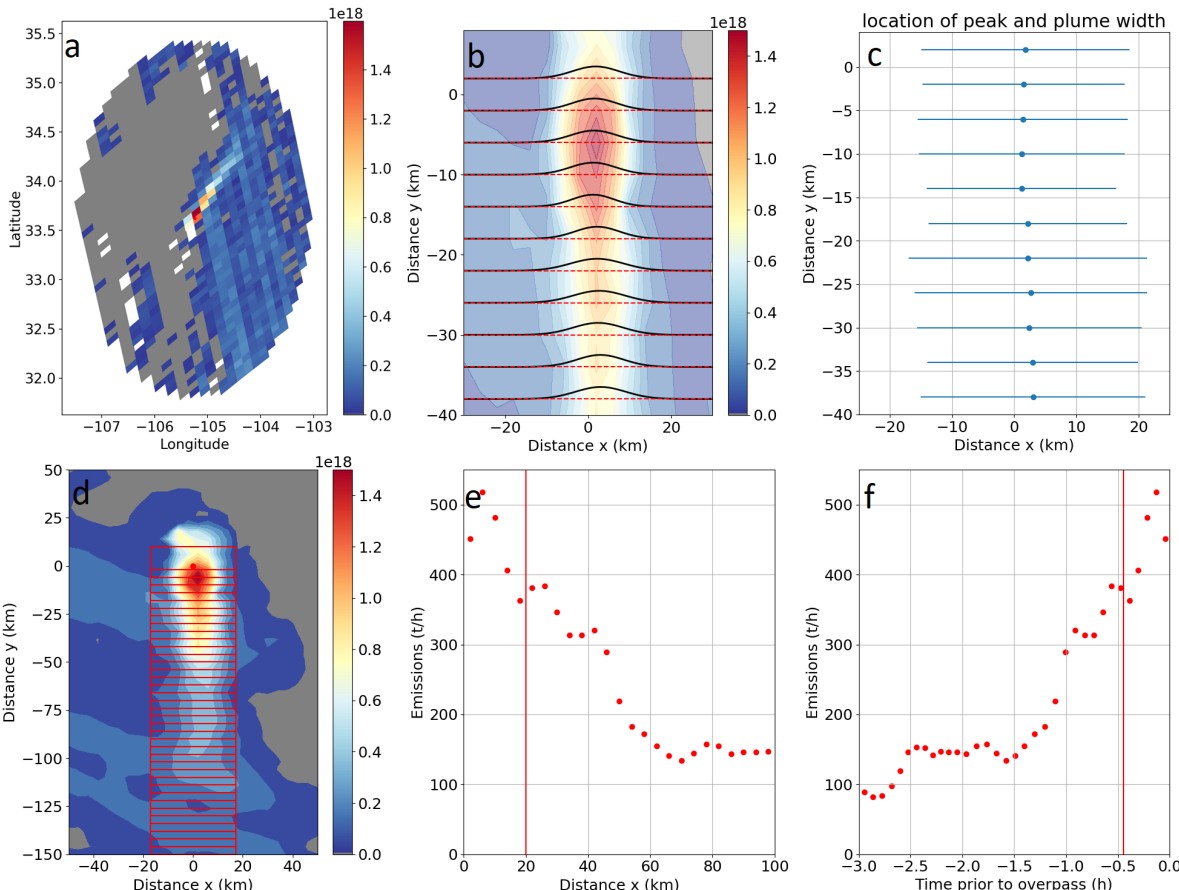

**Figure 1.** Illustration of the method to derive CO emissions: (a) the unmodified TROPOMI CO VCDs in a longitude-latitude domain, (b) simple Gaussians are fitted across wind in 4-km wide boxes up to 40 km downwind of the fire to find the plume width, correct the wind direction and fire location, (c) the peak $x_0$ (blue dot) and 3-$\sigma$ (blue bars) from the Gaussian fit are used to find the plume width and correct the wind, (d) the VCDs are rotated with the corrected wind direction and the VCDs are integrated in boxes of 4-km by 3-$\sigma$, (e) the wind speed is applied to find the emission fluxes downwind of the fire, and (f) shows the same as (e) but projected in time since emissions occurred.

of the fire from the satellite observations of CO VCD (Fig. 1 a). Since the grid size is slightly smaller than the TROPOMI pixels the satellite observations are weighted by the actual pixel size and over-sampled by 7 km (Adams et al., 2019). Then, along the binned VCDs, Gaussian distributions are fitted across the plume in all downwind boxes (Figs. 1 b,c), to aligned the wind direction and determine the extend of the plume. The third standard deviation out from the plume center (i.e. 99 % confidence limit, Fig. 1 c) is used to define the plume lateral boundaries (typically between 10-30 km), and the wind direction correction is found by fitting a linear function through the centre of the peaks (Fig. 1 c). This new and corrected wind direction is then used to rotate the observations around the fire centre again (step 1 is repeated with the corrected wind direction). This approach is able to define the smoke plume in an automated way. A series of flux boxes of dimensions 4km in the wind direction, and 3$\sigma$ in

the direction perpendicular to the wind are superimposed on the binned VCD image (Fig. 1 d). Making use of the wind fields and VCD field, the flux is then calculated (Fig. 1 e), following Mebust et al. (2011)):

$$E_y = \sum (\Delta VCD) \times u \times A, \tag{2}$$

Where $\Delta$VCD are the flux differences with background levels removed, u is the horizontal wind velocity, and A is the area of the box. Background CO levels are taken from averages 20 to 50 km upwind of the fire. As recommended by Griffin et al. (2021), and based on model simulation tests (in Sect. 2.3), we define the average upwind concentrations as this CO background concentration. The grey areas in Fig. 1(a,d) thus correspond to concentrations at or below this background concentration level.

Individual box fluxes can be used to provide estimates of the source CO emissions several hours prior to the overpass. This also provides insight into the diurnal variability of fire emissions. To be successful, meteorological conditions must have been stable several hours prior to the overpass. Figure 1 shows a very good example of such a plume for which the diurnal variability could be determined; however, this is not the topic of this study, since its approach for estimating emissions several hours in the past is currently not automated or fully validated. For our final source emission estimate we use boxes within the first 20 km downwind of the fire to ensure that the time of the emission is close to the time of the overpass thus less influenced by the diurnal pattern. Since the average wind speed is roughly 20 km/h it is expected that the fire emission algorithm provides emission rates within 1 h of the overpass time. Thus, the time at which the emissions were released is expected to be approximately -30±30 min from the time of the TROPOMI overpass. Due to diurnal variability of fire emissions, the emissions estimate from this algorithm are time specific and do not represent a daily average. Any comparisons to emissions from other observations or inventories need to be made for the same time period. (Schaaf and Wang, 2015)

For input parameters, we utilize the wind fields (U, V) from the European Centre for Medium-Range Weather Forecasts (ECMWF) ERA5 reanalysis dataset at a resolution of $0.25° \times 0.25°$ with an hourly output, between 1000 and 300 hPa at a resolution of 50 hPa, and interpolated spatio-temporally to the TROPOMI observations. For large fires, the rotation of observations around a single point will cause imperfections, as they are not true point sources but are spread over large areas. However, the flux methodology captures the width of the plume in these cases, with the main effect the addition of some variability in the emissions at the first box of the overpass. To find the appropriate wind speed to use for emission transport, we use the average TROPOMI aerosol height (AER_LH) for each fire, which is a good proxy for the average height of fire plumes (Griffin et al., 2019). If there are no good quality plume heights near the fire, we use 2 km (or 800 hPa) (Griffin et al., 2020) for the plume height. This approach to find appropriate altitudes for wind fields has previously been successfully used to improve the accuracy of satellite-derived $NO_x$ emissions from biomass burning (Griffin et al., 2021).

As this method assumes steady state and relies on stable meteorology appropriate quality criteria need to be applied to filter any cases when the emission estimate might be deficient. The quality criteria to filter emission estimates as well as the total uncertainty of the emission estimates are further examined in the next section.

## 2.3 Accuracy of the emission estimates using synthetic data

Similar to Griffin et al. (2021), we conducted a sensitivity test using a regional air-quality model to create synthetic CO VCDs. Applying the fitting method used for satellite observations (as described in the previous section), source emissions retrieved from plumes generated within a model domain can then be compared directed to the original source emissions used by the model. For these tests all emissions are known, thus allowing us to: 1) test if the fitting method is able to regenerate the original emissions; 2) obtain a better idea of the uncertainties of the method; and 3) examine the extent to which quality filters should be applied to the satellite-derived emissions. For this sensitivity test, we use the Global Environmental Multiscale - Modelling Air-quality and CHemistry (GEM-MACH; Makar et al. (2015b, a); Chen et al. (2019)) air quality model to obtain the synthetic VCDs. The operational version of GEM-MACH that employs biomass burning emissions using CFFEPS (Chen et al., 2019), was used. This has a $10 \times 10\,\text{km}^2$ grid cell size for the North American domain, and 80 vertical levels (from the surface to approximately 0.1 hPa), further details can be found in Makar et al. (2015b, a). Although GEM-MACH's resolution is coarser than the 4 km pixel size used here, the higher pixel resolution becomes available as the pixels are binned using a distance weighted average. GEM-MACH provides hourly output, with an internal "physics" time step of 7.5 min. The meteorological component of GEM-MACH is within the physics module of the Global Environmental Multiscale (GEM) weather forecast model (Côté et al., 1998; Girard et al., 2014). GEM-MACH contains a detailed atmospheric chemistry scheme, which includes the emission and removal processes of 42 gaseous species and 8 particle species. The operational model run is initialized every 12 hours, at 00 and 12 UTC. Original input fire emissions are estimated based on hotspot location using the CFFEPS (Chen et al., 2019)), which links the hotspot locations to ecozone-specific databases of fire area per hotspot per unit time, fire stage (crown, duff layer, residual), and estimates plume height using plume rise calculations based on meteorological lapse rates and similar considerations. For a time period between May to September 2019, the model CO profiles are integrated over the first 39 layers (approximately the lowest 10 km of the atmosphere) to obtain VCDs over the model domain, in North America (Canada and US). The wind speed and wind direction used in this sensitivity test are based on forecast winds that drive the model simulations. The wind altitude for the synthetic retrieval is from the nearest model wind level to the predicted the aerosol layer peak concentration. For this time period (May to September 2019,in the US and Canada) a total emissions of 208 fires were "successfully" retrieved (a solution was found by the fitting algorithm). The results of all the retrieved emissions using the GEM-MACH output at 20UTC versus the original (synthetic) source are illustrated in Fig. 2 a. While the majority of the retrieved emissions is in close agreement with the original values (and are following the 1-to-1 line), there are noticeable outliers (about 10 out of 105), where the retrieved values are below the original emissions. An underlying requirement of the fitting methodology is the assumption that a steady-state in the meteorological conditions has been maintained during the time of the retrieval – previous work has shown that when this assumption is incorrect, the retrieved emissions may be in error (Fathi et al., 2021). Changes in meteorological state during the retrieval period (such as a change in wind direction or speed, changes in atmospheric stability) may influence retrieval accuracy, as may the presence of other sources nearby to the fire of interest. Examining these cases more closely we identified certain unfavorable conditions as follows (the filtered values due to these specific conditions are illustrated in Fig. 2:

1. The background ("B") is too high ($>0.7\times10^{19}$ molec/cm$^2$): this indicates potentially upwind sources, of large enough magnitude that the plume may be difficult to distinguish, or a misplacement of the fire centre, and thus these cases should be filtered (two estimates were filtered that way).

2. The variation in the wind direction ("wd") should to be less than 15°: changes in wind direction during the retrieval allow for potential convergence / divergence to occur within the model grid cell, violating the requirement of steady-state flow (78 estimates were filtered that way).

3. The plume width ("width") should be no larger than 50 km: the method cannot be used for very large fires, as the assumption of a point source breaks down; a width larger than 50 km could also be associated with interference from other nearby other sources (five estimates were filtered that way).

4. The difference of the emissions from individual cross-sections ($\Delta$ xsect) considered for the estimate (within 20 km of the fire) should be no larger than 100 %, and cases are also filtered where only one cross-section is used for the estimate: cases with high variability of the individual cross-sections indicate high variability of emissions within a very short time frame or unstable conditions (30 estimates were filtered that way).

Other parameters were also tested but were not included as part of the quality filter, such as variation in wind speed, maximum and minimum wind speed, height of the aerosol layer, the Richardson number and the wind shear. It should be noted that for all cases the minimum wind speed was above 2 m/s and the maximum wind speed was 11 m/s. We would expect that the method is not reliable for very high or very low wind speeds (approximately $>2$ m/s), as found by other studies (e.g. de Foy et al., 2014). After applying the quality filter (black points in Fig. 2) 105 fires remain (some filters overlap). The correlation is high between the retrieved and original source emissions with $R = 0.92$ and a slope of best-fit (using geometrical mean) of 1.1. The relative difference is 34 % (fitted-input) which is used in our uncertainty analysis (Table 1).

The four established quality controls noted above were then used to filter the satellite-derived emissions estimates, and are recommended in retrievals of this nature. In addition, for the satellite-derived emissions, a filter that requires at least five observations for the estimate has also been applied. These tests using synthetic data can also help to establish the uncertainties for the estimated emissions. The total uncertainty of the satellite-derived emissions (see Table 1), is based on the systematic bias and random uncertainty. The random uncertainties consist of the wind speed ($\approx 10$ %), the effect of the altitude used for the wind speed ($\approx 20$ %), and the uncertainty of the method itself (based on the relative difference between the true and fitted emissions of 34 % after applying the above mentioned quality filters). The uncertainty of the wind speed caused by the uncertain altitude of the plume is based on the mean difference of the wind speed when comparing the winds 50 hPa above and below the aerosol layer height. The uncertainty of the wind speed is based on Gualtieri (2022) who found approximately 0.5 m/s for the 90 % confidence interval for ERA5, with the average wind speed of approximately 5 m/s (for our dataset), we assume a 10 % uncertainty for the wind speed. These errors are added in quadrature, leading to a total random uncertainty of 41 %. Additionally, the TROPOMI CO VCDs (comparison to TCCON) are biased high by about 10 % (Sha et al., 2021), and

**Table 1.** Summary of uncertainties for the satellite emission estimates.

| Type | Uncertainty | Type |
|---|---|---|
| Method | 34 % | Random |
| Wind | 10 %[*] | Random |
| Wind altitude | 20 % | Random |
| Averaging Kernel | +6 %[**] | Systematic |
| Satellite VCD | +10 %[***] | Systematic |
| Total Random | 41 % | |
| Total | 57 % | |

[*] Gualtieri (2022), [**] Rowe et al. (2022), [***] Sha et al. (2021)

not accounting for the averaging kernel correction due to the lack of profile observations for the fires will add another 6 % (Rowe et al., 2022) (Table 3, difference between accounting and neglecting the averaging kernel). While the total emissions (or VCDs) could be scaled by the systematic bias, the effect of the averaging kernel correction depends on the profile as well as averaging kernel shape (also see averaging kernel analysis in Sect. 2.1 and Fig. B1). Adding the systematic and random error leads to a total uncertainty of 57 %.

Overall, the sensitivity tests suggest that the fitting method is robust once filters have been applied to ensure that the underlying assumptions of steady-state meteorological conditions is maintained for the observed data, and can be used to estimate the CO fire emissions. The total uncertainty of the CO emission estimates (after the above mentioned filters have been applied) is approximately 41% (random) and 57 % (total) based on the uncertainty of the wind speed, CO VCDs and methodology. Throughout Sects. 4 and 3 these same filters that ensure steady-state meteorological conditions and low interference from nearby sources were applied to the satellite-derived emission estimates.

### 2.4 Satellite FRP and Hotspot Identification

To find the locations of fires around the globe we use MODIS instrument thermal anomalies and FRP products. MODIS was used in this study for two purposes: (1) to obtain the fire locations and fire centres using MODIS thermal anomalies that are then used to attempt deriving CO emissions from TROPOMI; (2) to obtain the FRP for each fire to determine the emissions coefficient (EC; see Sect. 4). The MODIS instruments, on board the NASA Earth Observation System Terra and Aqua satellites, detect fires using data collected in the infrared and spectral channels (Kaufman et al., 1998). Typical overpass times occur at approximately 10:30 AM/PM and 1:30 AM/PM local time for the TERRA and AQUA platforms of which MODIS is a component, respectively.

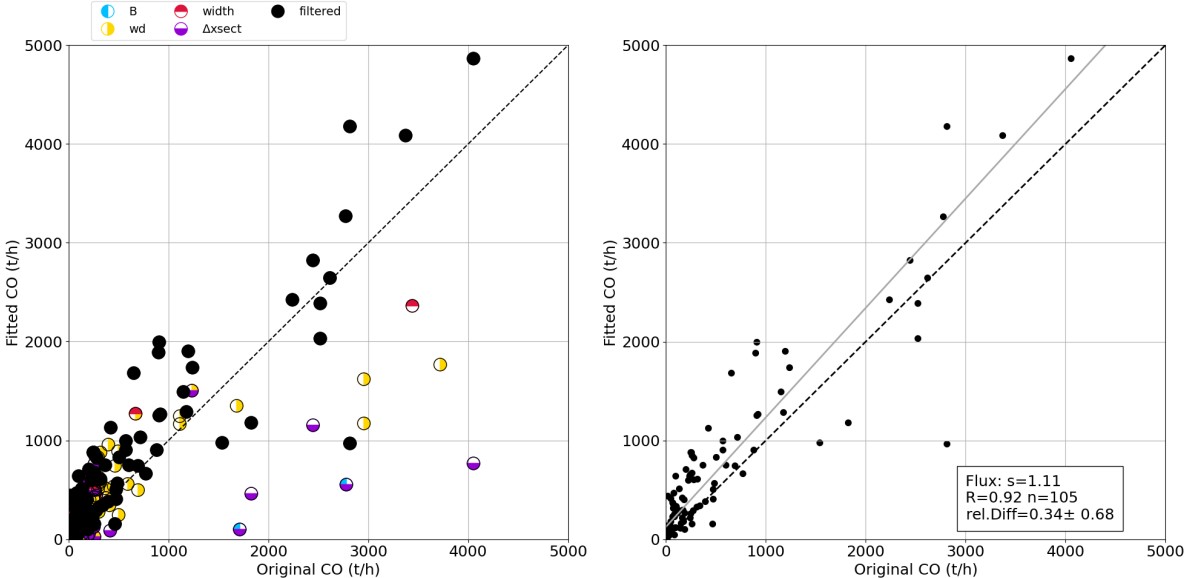

**Figure 2.** The results of the sensitivity test with synthetic VCDs are illustrated (see text for additional details). The filtering low quality results are illustrated in panel (a), where different parameters have been tested, including the maximum value of the background ("B"), the deviation of the wind direction ("wd"), the maximum width of the plume ("width"), and the difference between the individual cross-sections ("Δxsect"). The filtered fitted emissions versus the model input emissions are plotted in panel (b) together with the statistics (slope of best-fit using the geometric mean, s; correlation coefficient, R; the number of points, n; and the mean and standard deviation of the relative difference, rel. Diff: fitted-input).

The MODIS thermal anomaly product (MOD14) (Giglio et al., 2003, 2006, 2016) from Aqua (13:30 local time) is used in this study to locate the fires. These thermal anomalies are clustered, with the criterion of a minimum summed FRP (within a 5 km radius) of 1 GW (note, this threshold is not for individual hotspots but for the fire cluster) and a confidence of at least 75 %

(for individual hotspots). These thresholds have been applied to remove fires that are too small, as the direct TROPOMI CO emission estimate very likely fails for very small hotspots and to reduce the influence of other (smaller) sources causing thermal anomalies (e.g. flares). Depending on the size of the fire, we aggregate on average 30 thermal anomalies, based on Freeborn et al. (2014) this is associated with a 6 % uncertainty of the FRP, much lower compared to the uncertainty of the CO emissions estimates (57 %). Again, we would like to highlight that not all fires are captured by satellites (including MODIS), fires can

potentially be missed for several reasons: if the FRP signal is too low (e.g. small fires), due to cloud cover, and under thick smoke plumes. These locations are used to attempt an automated CO emission estimate with TROPOMI, however, it is not always possible to derive emissions from the fire hotspot location and many locations will fail or are filtered (as mentioned in Sect. 2.3) after the emission estimate. For a typical year using the described MODIS clustering, we are left with approximately 13-18 thousand fire clusters globally for which we attempt an emission estimate. For about 3-4 thousand fires the estimate fails

entirely. And further 9-12 thousand fire emissions are filtered due to poor quality, leaving 4-6 thousand successful fire emis-

sions globally per year. The most common reasons for failed or filtered emission estimates include variables winds, low CO columns that are too close to the background concentrations, nearby CO sources (such as a second fire plume), and cloud cover.

MODIS FRP is also used to estimate the ECs, presented in Sects. 3. For our CO emission inventory we use our estimated $EC_{CO}$ (Sect. 4) and apply these to the assimilated daily GFAS FRP on a $0.1 \times 0.1°$ grid to estimate a global inventory of CO emissions. GFAS is a top-down emission estimation system from ECMWF. The GFAS assimilated FRP is based on the MODIS Aqua and Terra FRP that provide typically one daytime and one nighttime overpass each. This dataset provides a guidance on total daily FRP that can then be combined with the derived ratio between TROPOMI CO emissions and MODIS FRP. Since TROPOMI only provides an emission estimate around 1:30PM (local time) accounting for diurnal variability is not feasible with TROPOMI alone. Thus, a secondary dataset such as MODIS, with multiple overpasses per day at various times, is necessary to get an approximation and of diurnal fire activity and ultimately to obtain a total emission inventory. This emission inventory is used in Sect. 5 to compare the here estimated CO emissions with fire emission inventories. It should be noted that some smaller fires might be below the MODIS detection limit, and will be missed and in the presence of clouds or thick smoke the instruments may not be able to observe the Earth's surface. The retrieved emissions generated here may therefore be lower limits.

## 2.5 Emission inventories

We compare our retrieved CO emissions to several existing biomass burning CO inventories, namely GFAS, GFED, FINN v1.5 and v2.5, as well as GFFEPS. GFED (van der Werf et al., 2017) and FINN are both based on bottom-up approach. Here, we use GFEDv4.1 which has a $0.25°$ resolution, developed by NASA, in Sect. 5 we use the annual total emissions for different geographical regions. The FINN inventory (Wiedinmyer et al., 2006, 2011) by NCAR is not on a regular grid but based on the location of the MODIS or VIIRS detected hot spots, these are then summed to obtain annual totals in Sect.5. The GFAS fire emission inventory (Kaiser et al., 2012) by ECMWF utilizes a top-down approach based on MODIS FRP and is on a $0.1°$ regular grid, here we use v1.2. Additionally, we also compare to a new global biomass burning algorithm, GFFEPS, developed by Environment and Climate Change Canada. GFFEPS is a global extension of the CFFEPS model described in Chen et al. (2019). Similar to CFFEPS, GFFEPS is a bottom-up approach utilizing satellite-detected hotspots to calculate smoke emissions. The resulting emissions are not gridded, but distributed to the location of the detected fire-hotspots. The model uses the Visible Infrared Imaging Radiometer Suite (VIIRS) that then predicts emissions based on the Canadian Forest Fire Danger Rating System (CFFDRS, Stocks et al. (1989)). Fuel types were assigned using the Global Land Cover (GLC) 2000 (European Commission, 2003). Area burned per hotspot was estimated based on eight years of satellite hotspot data (2012-2019) and reported area-burned statistics for the same time periods and locations using MCD64CMQ (Giglio et al., 2020). Daily fire weather conditions based on the Canadian Forest Fire Weather Index (FWI) System (van Wagner, 1987) were calculated in the global version of ECCC's Global Environmental Multiscale (GEM) model (Côté et al., 1998) and interpolated to hotspot locations. Fire behaviour conditions at each hotspot were based on the Canadian Forest Fire Behaviour (FBP) System (Forestry Canada Fire Danger Group, 1992) and calculated in the Canadian Wildland Fire Information System (Lee et al., 2002)

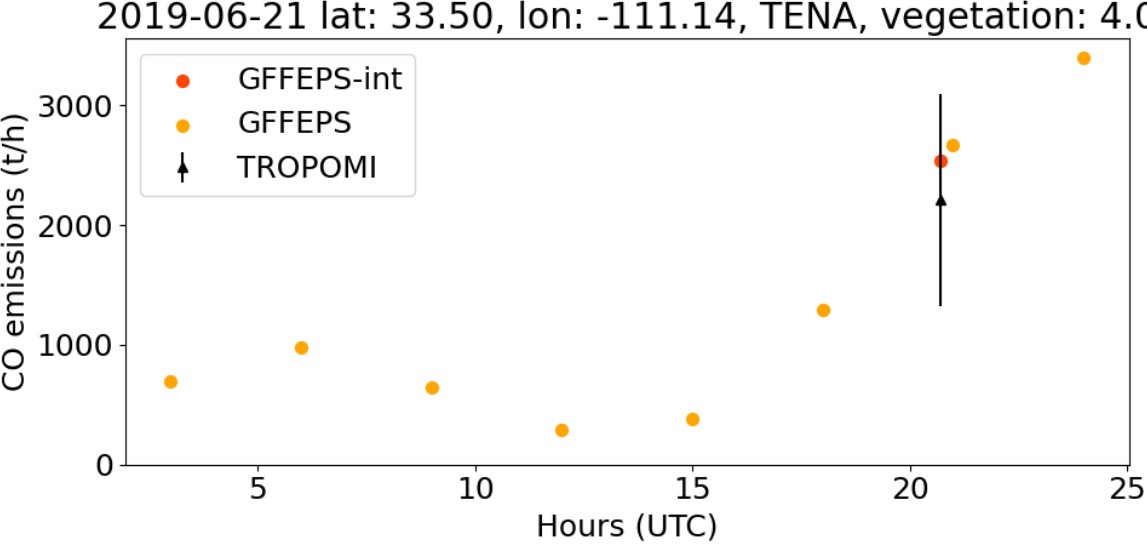

**Figure 3.** Comparison between TROPOMI-derived CO (black triangle), and GFFEPS (yellow) CO emissions on the example of a fire in Arizona, US (TENA, 33.5°N, 111.14°W) on 21 June 2019. Also shown are the time-interpolated GFFEPS emissions to the time of the TROPOMI overpass (red, "GFFEPS-int").

operated by the Canadian Forest Service, Natural Resources Canada (https://cwfis.cfs.nrcan.gc.ca/; last access: Feb. 7, 2023). Surface and crown fuel consumption rates are translated directly into smoke emissions. Emissions rates per species per stage of combustion are based on Urbanski (2014). A fix diurnal profile is applied to the daily estimated burn area to obtain an hourly fraction with peak activity at 5 pm local time. Note that assignment of GLC 2000 land classifications to Canadian Fuel types and adjustments to fit global conditions continues to be an area of development in the model. Fuel loads were largely taken from van Leeuwen et al. (2014) and van der Werf et al. (2017) as used in GFED.

## 3   Evaluation of direct vs bottom-up emissions

CO fire emission can be estimated from TROPOMI single overpass observations. Stockwell et al. (2022) have shown good agreement between the TROPOMI-derived to aircraft-derived CO fire emissions as part of the FIREX-AQ campaign (Warneke et al., 2023). The sensitivity tests (Sect. 2.3) using synthetic total columns also suggest that emissions can be reliably estimated using the flux method within 57 % uncertainty. In this section, TROPOMI-derived emission estimates are used to evaluate the GFFEPS emissions processing system. Figure 3 shows an example of a fire in Arizona on June 21, 2019 in the Temperate Forest North America (TENA) region (33.5°N, 111.14°W). GFFEPS emissions are given in 3 h intervals and shown as orange dots. GFFEPS estimates daily emissions based on area burned, and utilizes a prescribed diurnal pattern with a peak in fire intensity and emissions in the late afternoon. The peak in emissions always occurs a few hours after the TROPOMI overpass.

TROPOMI overpass time was around 20:30 UTC and the emission estimate is shown as a black triangle. The GFFEPS emissions are interpolated to the TROPOMI overpass time (shown as a red dot), and for this example GFFEPS aligns with the satellite-derived emissions very well.

The TROPOMI-derived emissions can be used more broadly to examine the performance of the GFFEPS emissions. Figure 4 (a) shows the comparison between TROPOMI and GFFEPS (at the overpass time of TROPOMI, equivalent to the red dot

in Fig 3) for roughly 4000 fires globally in 2019 (all fires where TROPOMI could successfully estimate CO fire emissions, appropriate filters as described in Sect. 2.3 have been applied). The results show that the model captures the order of magnitude and some of the variability, however, on average GFFEPS tends to predict lower emissions than the satellite-derived CO emissions. This discrepancy is likely due to an underestimation from GFFEPS rather than an overestimation of TROPOMI emissions: TROPOMI offers high-quality data over fires and smoke plumes (see Fig 5) and the effect of applying an averaging

kernel correction would lead to even higher emissions (see Sect. 2.1 and Fig. B1). It is important to note that the TROPOMI CO emissions approach has been validated (Stockwell et al., 2022) and has a 57 % total uncertainty see Sect. 2.3), while GFFEPS still requires validation and associated uncertainty estimates.

The cause of the differences between the TROPOMI-derived and GFFEPS emissions are being investigated; possible rea-

365 sons for this could be the following: 1) misrepresentation of fuel type and/or its associated emissions factors; 2) the estimated area burned could incorrect; 3) the diurnal variability is not accurately represented; or 4) the time assumed for the TROPOMI-derived emissions is not correctly represented. To examine the reasons more closely and to pinpoint the issues, specific areas and fuel types were examined individually (see Tables D1 and D2). The region classification relies on the definition used in GFED (Giglio et al., 2003), which divides the world into 14 distinct areas: boreal North America (BONA), temperate North

America (TENA), Central America (CEAM), Northern Hemisphere South America (NHSA), Southern Hemisphere South America (SHSA), Europe (EURO), Middle East (MIDE), Northern Hemisphere Africa (NHAF), Southern Hemisphere Africa (SHAF), boreal Asia (BOAS), Central Asia (CEAS), Southeast Asia (SEAS), equatorial Asia (EQAS), and Australia and New Zealand (AUST). The extend of these regions is illustrated in Fig. C1. Considering factors like slope, R (correlation coefficient), and RMSE (root mean square error), the model demonstrates strong agreement with satellite-derived emissions for specific re-

gions, namely CEAM, NHSA, EURO, and MIDE (see Table D1). Other regions, like AUST have a very poor correlation, slope and RMSE, indicating a need to improve the modelling of that region, which is currently still under development, such as improving the emission factors, correcting the fuel consumption and combustion completeness for eucalyptus (Anderson et al., 2024). In terms of biomes (see Table D2), the results are less clear, as biome 1 dominates the AUST region for fires in 2019 and also shows a very poor correlation.

We also examine individual fires contributing to this issue in Figure 5 which depicts an example of a fire where the GFFEPS and TROPOMI values compare well (top row of panels) and a fire where the GFFEPS values are much lower than the satellite observations. Shown are the TROPOMI CO VCDs (Figs. 5 a and d), the GEM-MACH VCDs (using GFFEPS emissions) in Figs. 5 b and e, and the true color image together with the MODIS hotspots (Figs. 5 c and f). The fires that tend to

385 be lower compared to the directly-derived CO emissions significantly are predominantly the ones that are influenced by thick smoke (and/or clouds). Some of the lower emissions from GFFEPS may suggest that fires emitting thick smoke may have underestimated hotspot values – a correction for fires influenced by thick smoke reducing the number of observable hotspots may be necessary.

Specifically at the overpass time, the emissions are often underestimated by GFFEPS when comparing individual fires (at the time of the TROPOMI overpass). The TROPOMI overpass time (1:30 pm local time) is exactly at a time when fires typically experience significant growth, one or two hours either side of the overpass time make a difference of approximately 30-50 % (see Fig. 3). Thus, the lower emissions could also be the result of a timing issue, either from GFFEPS or the time assumed for the TROPOMI-derived emissions. GFFEPS prescribes a diurnal emission profile, whereas the TROPOMI emissions provide an emission rate specific to a satellite overpass time. As can be seen in Fig. 3, the diurnal variation in GFFEPS emissions can be substantial, more than a factor of ten between peak and minimum. While the satellite data only allows evaluation of GFFEPS at the overpass time (and hence an evaluation over all times of emissions is not possible). We can conclude that model emissions (specifically at the overpass time) capture at least some of the variation in CO emissions at overpass time ($R = 0.22$), but are general biased low compared to the satellite derived emissions in the early afternoon. The total daily emissions and the diurnal
variability are still a large uncertainty, and cannot be readily evaluated using polar orbiting satellites such as TROPOMI alone (since TROPOMI can only provide limited times to obtain CO fire emissions).

Overall, TROPOMI CO emission estimates can be used to help with the evaluation of the emission model and help pinpoint certain areas that need further improvement.

**4  Emission Coefficients for Different Vegetation Types from Satellite-Derived Emissions**

The TROPOMI-derived CO emissions alone cannot be used to obtain annual total emissions because: 1) emissions of many fires will not be possible to be derived directly (e.g. due to unfavorable meteorology, cloud cover, size of the fire etc.); and 2) TROPOMI is in a low earth orbit observing each location once or twice per day and the TROPOMI-derived emissions are limited to the time of these overpasses. Therefore, to obtain annual total emissions to be able to compare this to other
fire inventories the FRP measurements from MODIS and MODIS-based gridded FRE (from GFAS) are used to obtain a TROPOMI/MODIS top-down inventory. Emission coefficients, here defined as the ratio between the direct TROPOMI-derived CO emissions and MODIS FRP, are applied to the FRE to obtain annual total emissions by region, and can then be compared to study trends over time (see Sect. 5). The emission coefficients are a measure of the burning efficiency of different vegetation types. The emissions coefficients can be determined from the correlation and slope of best-fit between the CO emissions and
coincident FRP observations (Mebust et al., 2011; Mebust and Cohen, 2014; Adams et al., 2019).

Emissions factors will change with different stages of a fire (flaming to smoldering); however, it is challenging to separate

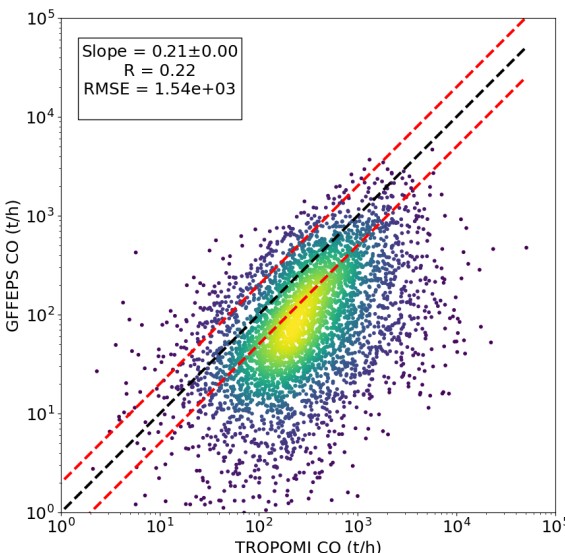

**Figure 4.** Comparison between approximately 4000 TROPOMI-derived CO emissions and (a) GFFEPS for 2019 fires. The colours indicate the density of the points (yellow being high density and blue being outliers). Note that the axis are in logarithmic scale, showing the 1-to-1 line (black) and the 1-to-2 lines (red). GFFEPS tends to be lower than the directly-derived TROPOMI emissions.

the different burning stages from fires (Andreae, 2019). For the emission coefficients derived in this study we did not separate the burning stages; instead, a single emission coefficient is used for each biome for the following reasons: 1) The MODIS-based GFAS FRE (total daily FRP) is a binned product, trying to project any assumptions of burning stages will introduce more uncertainty; 2) fires will most likely be flaming at the time of the TROPOMI overpass (1:30 pm local time). It is likely that these different burning stages have different CO emission coefficients (EC). For example, Hayden et al. (2022) identified that the CO EC is almost twice that during smoldering compared to flaming (this does not mean that emissions might be underestimated by the same amount, the FRP is significantly lower during smoldering stages and thus has a smaller effect on the total emissions). In that case study it was possible to roughly differentiate between smoldering and flaming; however, for large sample of fires it is very difficult to do so. The point is, that a mix of flaming and smoldering fires will reduce the correlation coefficient (e.g. Fig. 6), the overall EC will result in an average of flaming and smoldering EC. When applying these ratios globally to binned MODIS FRE to obtain annual emissions it is likely to average out overall, but all information on smoldering and flaming is lost in these averages. It is the norm for top-down inventories to apply a single emission coefficient (per vegetation) that does not change with the time of day (e.g. GFAS).

To differentiate biomass burning emissions in different biomes we use the GLC2000 (European Commission, 2003) as used for the development of GFFEPS, additionally we also tested the classification from MODIS (MCD12C1) and the GFED partition-

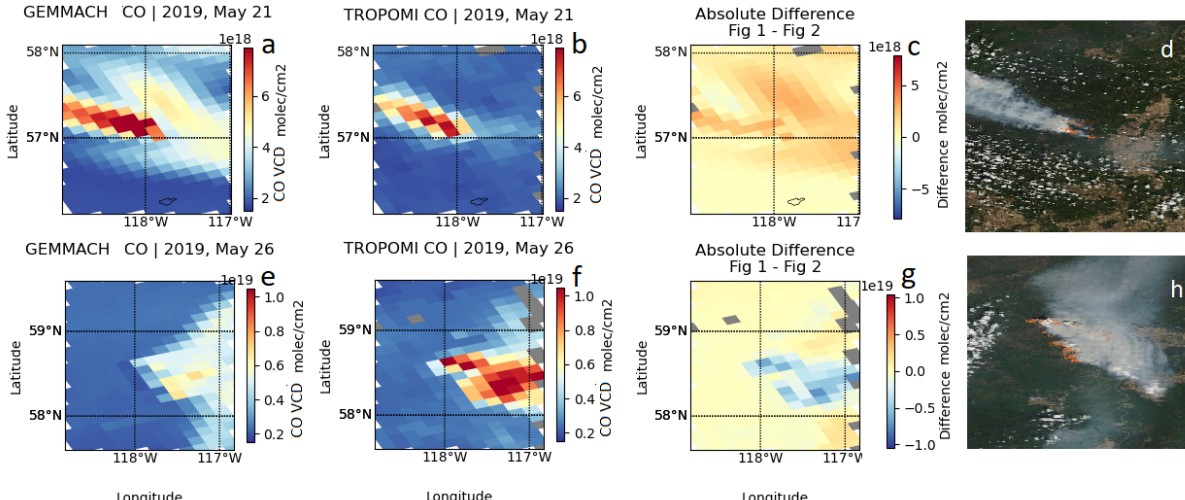

**Figure 5.** Comparison between model and TROPOMI-observed CO VCDs for two different fires. Top shows an example of a good match between the model (a) and TROPOMI (b), where the model input emissions (GFFEPS) are similar to the satellite-derived emissions. The difference (model−TROPOMI) can be seen in (c). (d) shows the VIIRS true color image for the same scene together with the day time MODIS thermal anomalies (hotspots, red dots), obtained from NASA Worldview; https://worldview.earthdata.nasa.gov/. The bottom figures show an example of a bad match between the model (e) and TROPOMI (f), where the model input emissions (GFFEPS) are significantly lower, with the difference shown in (g) and the VIIRS true color image in (h).

ing. The land use classification dataset distinguishes between 22 different types of biomes on 1 km resolution. Further details,
including the extend and location can be found in the Appendix, Fig. C2 and Table C1. In summary: 1-10 are different types of forests; 11-12 are types of shrub; 13-15 different types of grassland (herbaceous cover); 16-18 different types of Mosaic (cultivated areas and crops); 19-22 are areas where fires are unlikely (including water, snow, and urban areas). From the 22 possible types of biomes, we include a total of 15 different biomes in our analysis, excluded are biomes where fires were not observed by TROPOMI (and therefore no information is available on the CO emissions), namely: regularly flooded tree cover
(7 and 8), previously burned tree cover (10), bare areas (19), water bodies (20), snow and ice (21), and artificial surfaces (22).

Figure 6 shows the correlation between the TROPOMI-derived CO emissions (2019-2021) and the total MODIS Aqua FRP (which has a similar orbit as TROPOMI) for the corresponding fire for Tree Cover, broadleaved, evergreen (biome 1). The slope between the CO emissions (in g/s) and the FRP (in MJ/s) is EC (in g/MJ), the values of which are shown in Table 2 (and
445 for 2019, 2020, and 2021 in Tables E1, E2, and E3, respectively). A geometric mean approach was used to find the slope of best fit and the 99 % confidence level (used as the uncertainty of the EC). The results for all biomes used in the analysis are summarized in Table 2. The sample size specifies the number of fires used for the regression analysis and the rank is number of importance with respect to the total annual FRP from GFAS with 1 being the biome contributing the most (for the 2019 base year). The total FRP identifies how much each of these different biomes contribute to the total annual emissions (Sect.

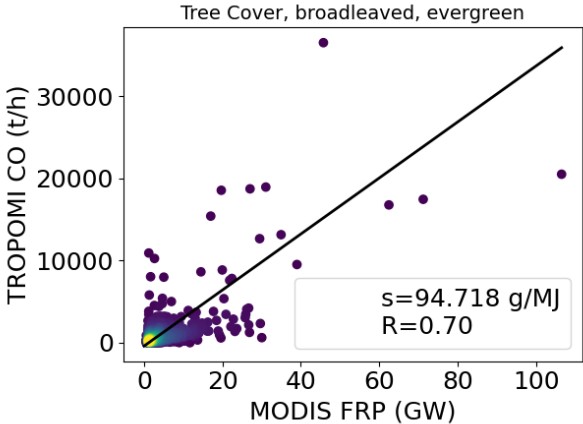

**Figure 6.** TROPOMI-derived CO emissions (2019-2021) versus MODIS detected FRP for fires from broadleaved evergreen trees. The color indicates the density (yellow being the most frequent and purple being single points) of the 842 fires. The black line indicated the slope ("s") of the best fit.

5). The emission coefficients vary between 111 and 38 g/MJ, where the largest CO emissions relative to FRP are from tree cover, broadleaved, evergreen (1) and the lowest are from shrub cover deciduous (12), meaning three times more CO is emitted from broadleaved evergreen trees (biome 1) compared to deciduous shrub (biome 12) for a fire that burns with equivalent heat energy. Evergreen shrub also has a high emission coefficient of 108g/MJ (biome 11) and even though both biome 11 and 12 are shrub, they are quite different biomes, based on their CO emissions and way they burn as well their location and occurrence.

Evergreen shrub (biome 11) is not very common (it covers approximately 0.5 % of the Earth's surface) and appears primarily in Central Asia and in some parts of Northern Canada and Alaska (see Fig. C2 and Table C1). Whereas deciduous shrub covers approximately 2.2 % of the Earth's surface and grows globally (see Fig. C2 and Table C1). Correlation coefficients vary significantly for different biomes. Most biomes have a moderate to high correlation with a correlation coefficient that is between approximately 0.5-0.9. The lowest correlation coefficient ($R = 0.39$) is found for Sparse Herbaceous/Shrub (biome 14) mean-

ing that the CO emissions are quite variable. A simplified classification of forest, shrub and grassland is not appropriate based on our results. For example the $EC_{CO}$ for different types of shrubs has both the largest and smallest emission coefficients and forests vary between roughly 38 and 108 g/MJ. An attempt was made to simplify the biomes following the approach of Mebust et al. (2011); however, this depreciated the correlation coefficients significantly.

The results using the MODIS MCD12C1 land classification instead are shown in Table 3. For water, snow/ice, urban, and sparsely vegetated no EC has been derived as there were unsurprisingly no fires found in the direct emissions from TROPOMI on these land types. As expected the EC for broadleaf evergreen forest are the highest (as for the GLC2000 definition) with 129g/MJ and the lowest are found for closed shrublands (23 g/MJ).

**Table 2.** Emissions coefficients for CO (EC$_{CO}$) ± the 99 % confidence interval (note this is only a mathematical error for the determination of the slope), correlation coefficient ($R$), number of fires (sample size) and rank of importance in terms of total FRP with 1 being the biome contributing the most to the emissions ("Rank"). All values shown are derived from TROPOMI and MODIS FRP for fires globally between 2019 and 2021 for a total of over 15,000 fires. The biome definition is taken from GLC2000, see Table C1.

| Number | Description | EC$_{CO}$ (g/MJ) | R | sample size | Rank |
|--------|-------------|------------------|---|-------------|------|
| 1 | Tree Cover, broadleaved, evergreen | 111±2 | 0.61 | 2459 | 1 |
| 2 | Tree Cover, broadleaved, deciduous, closed | 87±2 | 0.53 | 1418 | 4 |
| 3 | Tree Cover, broadleaved, deciduous, open | 58±1 | 0.56 | 2596 | 2 |
| 4 | Tree Cover, needle-leaved, evergreen | 57±2 | 0.49 | 832 | 8 |
| 5 | Tree Cover, needle-leaved, deciduous | 94±2 | 0.52 | 1987 | 10 |
| 6 | Tree Cover, mixed leaf type | 62±6 | 0.66 | 80 | 15 |
| 9 | Mosaic: Tree cover / Other natural vegetation | 64±3 | 0.71 | 346 | 12 |
| 11 | Shrub Cover, closed-open, evergreen | 108±4 | 0.64 | 452 | 13 |
| 12 | Shrub Cover, closed-open, deciduous | 38±1 | 0.62 | 1798 | 3 |
| 13 | Herbaceous Cover, closed-open | 45±2 | 0.47 | 878 | 6 |
| 14 | Sparse Herbaceous or sparse Shrub Cover | 45±3 | 0.39 | 363 | 9 |
| 15 | Regularly flooded Shrub and/or Herbaceous Cover | 101±5 | 0.61 | 357 | 14 |
| 16 | Cultivated and managed areas | 59±2 | 0.57 | 630 | 5 |
| 17 | Mosaic: Cropland / Tree Cover / Other natural vegetation | 62±3 | 0.40 | 464 | 7 |
| 18 | Mosaic: Cropland / Shrub or Grass Cover | 69±3 | 0.66 | 398 | 11 |

GFED relies on only 6 different types of vegetation (AGRI: Agricultural waste burning; BORF: Boreal forest fires; DEFO: Tropical deforestation and degradation; PEAT: Peat fires; SAVA:Savanna, grassland, and shrubland fires; and TEMF: Temperate forest fires) and the results of the emission coefficients using the TROPOMI direct emissions estimates (and MODIS FRP) are shown in Table 4. This gives the opportunity to be able to compare with the emission factors from GFED and GFAS directly (shown in Table F1 although the conversion factor to convert the ECs to EFs can change by a factor of four depending on the study used (Wooster et al., 2005; Kaiser et al., 2012), limiting a meaningful comparison). From our analysis: PEAT the highest ECs (183 g/MJ), and the lowest are found for agricultural waste burning.

There are certain advantages to combining the TROPOMI-derived ECs for CO and the MODIS FRP. It is possible with this approach to obtain an inventory of annual (or monthly) CO emissions from wild fires, because of the more continuous and diurnal coverage of MODIS FRP with at least 4 overpasses at various times during day and night (GFAS FRP). This can then be used to make a comparison between emission inventories and TROPOMI derived CO. There is also the potential to apply theses ECs to other years (even before the launch of TROPOMI), the following section will discuss this in more detail.

**Table 3.** Emissions coefficients for CO ($EC_{CO}$) ± the 99 % confidence interval (note this is only a mathematical error for the determination of the slope), correlation coefficient ($R$), number of fires (sample size) and rank of importance in terms of total FRP with 1 being the biome contributing the most to the emissions ("Rank"). The values are derived from TROPOMI and MODIS FRP for fires globally between 2019 and 2021 for a total of over 15,000 fires. The biome definition is taken from MODIS MCD12C1.

| Description | $EC_{CO}$ (g/MJ) | R | sample size | Rank |
|---|---|---|---|---|
| Evergreen needleleaf forests | 76±4 | 0.61 | 246 | 9 |
| Evergreen broadleaf forests | 129±3 | 0.61 | 1903 | 4 |
| Deciduous needleleaf forests | 69±7 | 0.48 | 100 | 12 |
| Deciduous broadleaf forests | 39± 2 | 0.42 | 425 | 7 |
| Mixed forests | 78±6 | 0.47 | 212 | 8 |
| Closed shrublands | 23±3 | 0.52 | 61 | 13 |
| Open shrublands | 44±2 | 0.32 | 718 | 6 |
| Woody savannas | 82± 1 | 0.59 | 2558 | 3 |
| Savannas | 77±1 | 0.48 | 6202 | 1 |
| Grasslands | 56±1 | 0.56 | 2486 | 2 |
| Permanent wetlands | 87±12 | 0.38 | 64 | 5 |
| Croplands | 37±2 | 0.58 | 353 | 10 |
| Mosaic | 43±4 | 0.92 | 23 | 11 |

**Table 4.** Emissions coefficients for CO ($EC_{CO}$) ± the 99 % confidence interval (note this is only a mathematical error for the determination of the slope), correlation coefficient ($R$), number of fires (sample size) and rank of importance in terms of total FRP with 1 being the biome contributing the most to the emissions ("Rank"). The values are derived from TROPOMI and MODIS FRP for fires globally between 2019 and 2021 for a total of over 15,000 fires. The biome definition is taken from GFEDv4.

| Description | $EF_{CO}$ (g/kg) | R | sample size | Rank |
|---|---|---|---|---|
| AGRI | 49±2 | 0.65 | 391 | 5 |
| BORF | 91±2 | 0.51 | 2845 | 3 |
| DEFO | 52±1 | 0.50 | 1381 | 2 |
| PEAT | 183±35 | 0.59 | 25 | 6 |
| SAVA | 60±1 | 0.53 | 9831 | 1 |
| TEMF | 110±4 | 0.60 | 764 | 4 |

## 5 Global CO total emissions

Applying TROPOMI-MODIS-derived ECs (from Sect. 4) to daily integrated MODIS FRE (that captures the diurnal fire activity) can help to obtain the total CO emissions and can allow to compare it to existing fire emission inventories. The ECs

(derived in Sect. 4) are here applied to the daily GFAS FRP, assimilated FRP observations from the Terra MODIS and Aqua MODIS satellite sensors. The domain is global with a resolution of $0.1°$ on a regular latitude-longitude grid. The time period between 2003 and the present is covered. The resulting top-down emissions, referred to as "TROPOMI-FRE", can be used to study the distribution of fire emissions more generically. Figure 7 shows the distribution of the CO emissions for 2019 by (a) biome, (b) region, and (c) month. Most CO emissions are from evergreen forests (biome type 1), which also has one of the

largest EC for CO (see Table 2). About two thirds of biomass burning CO emissions are from forests (GLC2000 type 1-6). The regions affected by the largest CO fire emissions are SHAF (26.5 %) and SHSA (16.8 %). The least affected are EURO and the MIDE, these regions have the least amount of fires and the smallest biomass burning CO emissions, below 1 % of the global emissions. The annual cycle is a little more evenly distributed with emissions between roughly 5-14 %, and shows the peak of CO emissions in August (14 %) with the lowest emissions in May (4.9 %). It should be noted that 2019 was an unusual

495  year for Australia with high intensity fires in December 2019 (during the 2019/2020 Australian summer), also known as the "black summer" (e.g. van der Velde et al., 2021; Pope et al., 2021). These fires contributed significantly to the global December emissions (Fig. 7 (c)). The type of biome burned was Eucalyptus forest, which is classified as biome 1.

The total biomass burning related CO emissions using TROPOMI-FRE are approximately 290 Mt in 2019. Using differ-
ent land classifications we get 308 Mt using the MCD12C1 classification and 265 Mt for GFED classification, these help to determine an overall uncertainty for the methodology and the total emissions of the TROPOMI-MODIS top-down emissions. Using the coarser classification of GFED leads to lower emissions ($\sim$ 15Mt) in the SHSA and SHAF region compared to using the ECs for GLC2000 or MCD12C1 classification. To assess the uncertainty of the total annual emissions of our estimates (TROPOMI-FRE), we also used emission coefficients derived from fires of individual years (2019 to 2021, using the GLC2000

classification). Using emission coefficients from 2019,2020, and 2019-2021 combined did not impact the total emissions for individual regions or globally (see Fig. E1). 2021 seems to be the anomaly for which the total global emissions reduced by approximately 20-25 % , due to overall lower $EC_{CO}$ (for biomes 1-3, see Table E3 which affected the SHAF region the most and almost halved the emissions in 2021 compared to using the ECs from other years). This shows that the uncertainty of our approach is at least 25 % for the global emissions but for individual regions an uncertainty of 50 % should be assumed. The

impact of using ECs from individual years is greater than the impact of using different definitions of biomes. It should be noted that the uncertainty discussed here is only due to the ECs, the uncertainty due to GFAS FRE (not provided) does not go into this estimate thus the total uncertainty of the TROPOMI/FRE top-down emissions is expected to be higher. For most biomass burning emissions inventories an uncertainty of a factor of two is assumed (Pan et al., 2020; Wiedinmyer et al., 2023).

The total from TROPOMI-FRE may be compared to those from other fire emission inventories: GFFEPS (337 Mt), the top-down inventory GFAS (364 Mt), the bottom-up inventories: GFED (408 Mt), FINN v1.5 (295 Mt), and FINN v2.5 (579 Mt). The break-down for 14 common geographical regions around the globe, as defined by (Giglio et al., 2013, Fig. 1), can be seen in Fig. 8. Even though the here presented TROPOMI-FRE product is based on GFAS FRP, there are significant differences between these two inventories. Most noticeable in EQAS and BOAS region where TROPOMI-FRE is significantly lower and

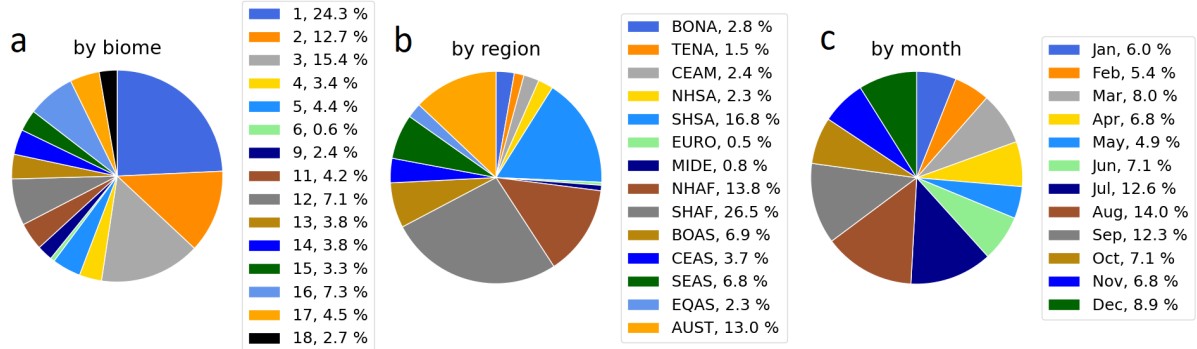

**Figure 7.** Analysis of the 2019 TROPOMI-FRE CO fire emissions: (a) emissions by different biomes as defined in Table C1, (b) by geographical regions from GLC2000 as defined by Giglio et al. (2013)), (c) by month.

SHAF region where it is higher. FINN v2.5 stands out as the fire emission inventory with the highest emissions, almost twice as large as FINN v1.5. FINN v2.5 emissions are especially high for SEAS. The inventories show consistently the largest CO emissions due to biomass burning from SHSA and SHAF (with most fires just south of the equator), a result of the Amazon tropical forest fires and Congo's forested ecosystem, respectively. For BONA, TENA, CEAM, NHSA, EURO, MIDE, NHAF, and CEAS, there is agreement within a factor of two (but often better than that) between the different inventories. EQAS and BOAS are the regions where CO emissions are the most variable and the least consistent between the different inventories with rates between approximately 10-80 and 10-70 Mt for 2019, respectively. The most noticeable differences are for FINN v1.5 and v2.5 that are exceptionally high in the SEAS region compared to all other estimates. CO emissions from EQAS and BOAS seem high for the GFED and GFAS estimates compared to the other inventories. BONA CO emissions are also about twice as high for GFED and GFAS compared to the other estimates. AUST is very low for FINN v1.5 (roughly a factor of five) compared to the others. The low bias in GFFEPS (as found in Sect. 3) cannot be seen in this comparison. One possibility is that the bias seen in the previous section is due to a inaccurate diurnal pattern, but there could be other reasons too. To be certain, further investigation is required that is outside the scope of this study. Overall, Fig. 8 highlights that there are large discrepancies especially for certain regions around the globe with regards to biomass burning related (CO) emissions. Using a measurement based approach can help with the evaluation of the different inventories, as was done in Sect. 3 for individual fires. Again, it should be highlighted that all these top-down and bottom-up inventories rely to a certain extend on good coverage of hotspot locations. If the fire hotspots cannot be measured by MODIS (due to clouds or thick smoke) the emissions will be underestimated, which is a difficult bias to properly correct for without introducing further assumptions and uncertainties. If anything, the true CO fire emissions are likely higher than the ones here presented, due to missed fire hotspots and the underestimate of large fires (with thick smoke).

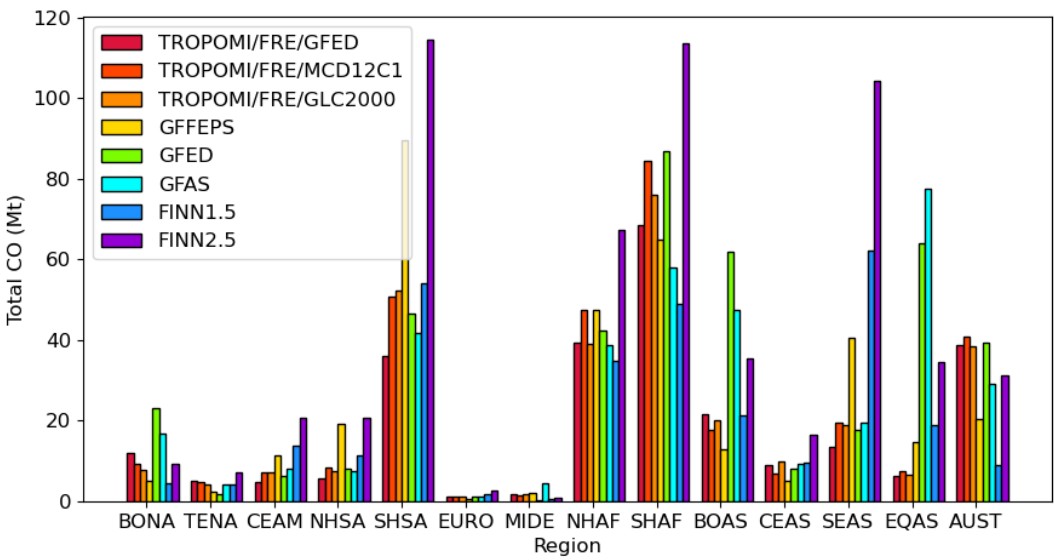

**Figure 8.** Comparison of total CO emissions from fires in 2019 for different fire emission inventories and TROPOMI-FRE based on different land classification for different geographical regions (as defined by Giglio et al. (2003).

## 5.1 CO emissions over the past two decades

The inventories discussed in the previous section provide data for various past years, except for GFFEPS (currently only available for 2019). For our independent estimates, we relied on daily FRP data from GFAS, which is based on MODIS FRP, available from 2003 to the present. Under the assumption that the $EC_{CO}$ values (as derived in Sect. 4) remain relatively stable over the years, we conducted an extensive analysis of the entire time series and calculated CO emissions spanning from 2003 to 2021 (refer to Fig. 10) using the GLC2000 classification for the ECs. Furthermore, we also present data from the other four inventories for the same time frame. The results are visualized in Fig. 9.

As expected, the emissions from biomass burning in various regions across the globe exhibit significant interannual variability. Notably, EURO and MIDE consistently report the lowest biomass burning emissions throughout the entire time series and are barely noticeable in the figures. The predominant source of biomass burning CO emissions is from SHAF and SHSA, followed by NHAF. This consistent pattern is evident for all the inventories analyzed.

To enhance the clarity of emissions identification and changes across different regions, we have depicted emissions by region in Fig. 10. The rate of change for this time period has been quantified for each inventory, and the results are presented in Table 5. Significant rates of change (with a p-value below 5 %) are highlighted in bold, while all other rates of change are statistically insignificant.

Globally, CO emissions are experiencing a decrease ranging from 5.1 to 8.7 Mt(CO)/yr between 2003 and 2021 across all inventories, with the exception of GFED. Notably, GFED does not reflect a global decrease due to the substantial increase in

CO emissions within the BOAS region, amounting to 19.8 Mt(CO)/yr. This overall decrease is primarily driven by significant reductions in SHSA (ranging from 2.1 to 6.3 Mt(CO)/yr), NHAF (ranging from 0.6 to 7.6 Mt(CO)/yr), SHAF (ranging from 0.9 to 5.6 Mt(CO)/yr), and CEAS (ranging from 0.3 to 3.3 Mt(CO)/yr), all of which show statistically significant decreases across at least four inventories.

In contrast, CO emissions from biomass burning are on the rise in TENA, with an increase ranging from 0.2 to 4.1 Mt(CO)/yr. Additionally, emissions in the EQAS region exhibit an interannual cycle that appears to correlate with El Niño years, resulting in higher emissions across all inventories in 2006, 2009, 2014, 2015, and 2019.

These findings align with prior research. Giglio et al. (2013) reported a decreasing trend in the annual area of land burned since 2000, which corroborates our observed reduction in CO emissions. Moreover, Zheng et al. (2021), also observed a decline in burned area between 1998 and 2015 through satellite observations, but reported stable or only slight decreases in biomass burning emissions. The satellite instrument "Measurement of Pollution in the Troposphere" (MOPITT) on board the TERRA satellite (Drummond et al., 2010) has been observing CO since 2000, (Buchholz et al., 2021) showed that MOPITT CO has been steadily decreasing by -0.50 % per year between 2002 to 2018. No study examining fire emissions for the time period presented here currently exists to our knowledge.

**Table 5.** Trends of CO emissions from wild fires (in units of Mt/yr) between 2003 and 2021 for 5 different fire emission inventories discussed in this study. Bold numbers indicate a significant trend (with p_value smaller than 0.05), for all others the trend is not significant. A negative number demonstrates declining emissions and a positive number indicates increasing emissions. Note that the uncertainty of these trends are large, which can be seen by the range of values across the different emission inventories.

| Region | TROPOMI-GFAS | GFED | GFAS | FINN1.5 | FINN2.5 |
|--------|--------------|------|------|---------|---------|
| global | **-5.3** | 0.4 | **-5.1** | **-8.2** | **-8.7** |
| BONA | 0.0 | 11.9 | -0.1 | -0.1 | 0.3 |
| TENA | **0.2** | **4.1** | **0.2** | 0.0 | **0.4** |
| CEAM | -0.1 | -1.2 | 0.0 | 0.0 | -0.5 |
| NHSA | 0.0 | 0.1 | 0.0 | 0.0 | -0.1 |
| SHSA | **-2.1** | -7.3 | **-2.2** | **-3.0** | **-6.3** |
| EURO | 0.0 | -0.2 | 0.0 | **0.0** | -0.1 |
| MIDE | **0.1** | 0.1 | **0.2** | 0.0 | **0.0** |
| NHAF | **-1.3** | **-7.6** | **-1.0** | **-0.6** | 0.5 |
| SHAF | **-1.1** | **-5.6** | **-0.9** | **-1.0** | 0.8 |
| BOAS | 0.2 | **19.8** | 0.1 | -0.8 | -0.2 |
| CEAS | **-0.3** | **-3.3** | **-0.4** | **-0.6** | **-0.7** |
| SEAS | **-0.5** | -1.7 | **-0.5** | -1.2 | -1.4 |
| EQAS | -0.2 | -6.7 | -0.4 | -0.8 | -1.6 |
| AUST | -0.1 | 1.6 | 0.0 | 0.1 | 0.1 |

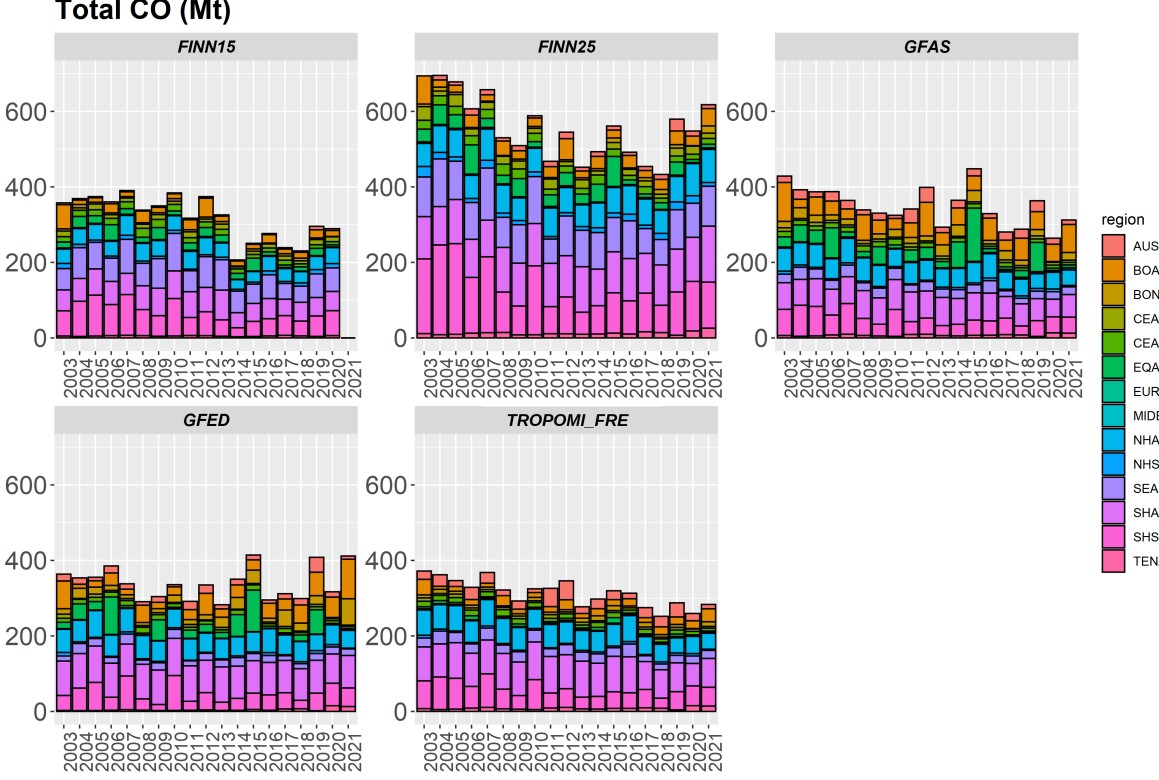

**Figure 9.** Total CO emissions from fires between 2003 and 2021 for different geographical regions (as defined by Giglio et al. (2003)) from FINN1.5, FINN2.5, GFED v4.1, GFAS and "TROPOMI-GFAS", which combines GFAS FRP and TROPOMI-MODIS EC$_{CO}$ (as defined in Table 5).

## 6 Conclusions

In this study, we presented an approach to compare TROPOMI direct-derived biomass burning CO emissions with those from bottom-up and top-down emission inventories and emission prediction systems. With TROPOMI CO observations, biomass burning emissions can be estimated from individual overpasses, resulting in roughly 5000 high quality fire emission estimates globally per year. The TROPOMI-derived CO estimates have preciously been validated with aircraft derived emission rates (Stockwell et al., 2022); here, we further established and automated an estimation method and performed model sensitivity tests to verify the robustness and uncertainties of the approach. Applying the same method to synthetic model VCDs, showed that the method is robust and is capable to derive the model input emissions when appropriate quality filters are applied. The success rate depends primarily on favorable meteorological conditions, including stable atmospheric conditions, low cloud and smoke cover, and no significant wind shear in the area, as well as the proximity of other nearby sources (especially upwind). Applying appropriate filtering for unfavorable wind conditions is the key for this method to work well and to reduce uncertain-

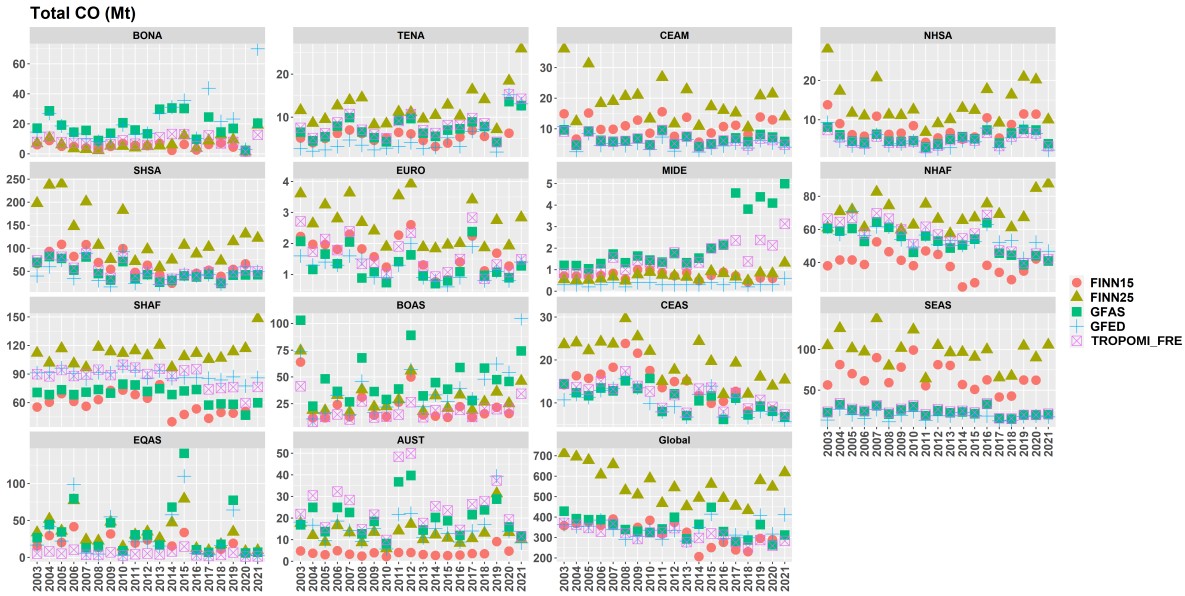

**Figure 10.** To illustrate the trend of CO fire emissions the sum of all 5 inventories (FINN1.5, FINN2.5, GFED v4.1, GFAS and "TROPOMI-GFAS") is shown for the different geographical regions between 2003 and 2021.

ties of the emission estimates. The sensitivity tests show that the methods uncertainty is approximately 34 % (and 57 % total uncertainty including the uncertainty of VCDs and winds).

Even though emissions for many fires are derived directly from TROPOMI CO, TROPOMI alone cannot be used to study total emissions, because too many fires are missed or filtered due to unfavorable meteorological conditions and the satellite is limited to a single daily overpass at 1:30pm. Thus, a top-down emissions inventory was obtained ("TROPOMI/FRE"): the TROPOMI directly-derived CO emissions have been used to obtain ECs with respect to MODIS FRP. These ECs are applied to GFAS FRE and result in total emissions. The TROPOMI directly-derived CO emissions have generally an average corre-

lation with FRP ($R \sim 0.5 - 0.6$) for most biomes, the slope (burning efficiency) varies by biome. In this study we used the GLC2000, MCD12C1, and GFED biome classification, and showed that there are large differences of EC for different types of forests. Based on this analysis we would not recommend a more simplified classification (e.g. for forests, shrub, grass). The TROPOMI-FRE top-down emissions uncertainties (based on the uncertainties of the ECs) are at least 50 % for regional emissions or 25 % for global emissions. Note that the total uncertainty is likely larger as the uncertainty of GFAS FRE (unknown)

is not accounted for.

We also found that the FRP is strongly influenced by thick smoke which can influence these types of top-down emission estimates, and leads to an underestimate of fire emissions for fires with thick smoke (typically large fires). The directly-derived

TROPOMI CO emission estimates are not impacted as much by the by the smoke and have been used here to verify and anal-
yse this issue for individual fires. This study highlighted the importance of hotspots: when these are obscured, fire emissions
cannot be estimated correctly with neither top-down or bottom-up that both rely on MODIS or VIIRS derived fire products.
Differences between flaming and smoldering are also expected and will influence the correlation between the CO emissions
and FRP, however, it is very difficult to determine the burning stage especially on a global scale for thousands of fires. When
estimating the ECs the smoldering and flaming stages are neglected. There is much uncertainty in this method as a single EC
is assumed for each biome that is based on emission estimates at around 1:30pm local time and thus the TROPOMI scenes in
general represent a small selection of conditions, it is unlikely to be representative of all fires or fire stages. The EC can also
be affected by sampling biases because the size of fires and the meteorological conditions that are selected. In the near future
geostationary satellites can be used to study total daily emissions and the ECs at different times of day.

In the comparison to GFFEPS, we identified the limitation to hotspot detection. GFFEPS relies on the satellite detected
hotspots, these feed directly into the estimate of burned area which is used to derive the bottom-up emissions. Thus, if not
all hotspots are captured the emissions will be underestimated. This discrepancy can be corrected for, but further analysis is
needed how exactly this can be accounted for.

The comparison of TROPOMI-FRP derived top-down emissions with other inventories reveals disparities and highlights the
substantial uncertainties associated with fire emission estimates. Notably, GFFEPS generally exhibits the closest agreement
with TROPOMI-FRE (e.g. total emissions for 2019: 337 vs 290Mt of CO for GFFEPS versus TROPOMI-FRE, respectively)
, although some exceptions are evident in regions such as SHSA, NHSA, SEAS, and AUST. Possible reasons for the discrep-
ancy are the omission of small fires, the absence of detected hotspots, and an underestimation of FRP, ultimately contributing
to an overall underestimation of total emissions. FINNv2.5 shows the largest discrepancies with TROPOMI-FRE where the
FINNv2.5 global emissions (579 Mt) are almost twice of TROPOMI-FRE, in the SEAS region the FINNv2.5 emissions are
higher by a factor of five: 20Mt versus 100 Mt).

Examining the trends over the past two decades (corresponding to the MODIS lifetime), it appears that global CO biomass
burning emissions have, on the whole, decreased, the trend (between -8.7 to -5.1 Mt/yr) is significant for all inventories, except
GFED (this is driven by the high increase for BOAS of almost 20Mt/yr). The trend of biomass burning emissions is highly
region-specific, with the highest reductions occurring in SHSA, SHAF, NHAF, and CEAS. Conversely, biomass burning emis-
sions in TENA are on the rise (by 0-4 Mt/yr). For all other regions, the variability within the past two decades has been too
substantial to determine a statistically significant trend.

Overall, directly-derived TROPOMI CO emissions and CO VCD enhancements are a great tool for validation of fire emission
models, e.g. GFFEPS. With this is was possible to pinpoint several issues, which either have been addressed or will be addressed
in future versions of GFFEPS, such as the obstructed hotspots. Geostationary satellite sensors, such as TEMPO (covering North

America), Geostationary Environment Monitoring Spectrometer (GEMS), or Sentinel-4 (covering Europe and Africa) will help to validate the diurnal pattern of emissions (for example for $NO_2$ and HCHO, note that none of the current geostationary satellites have the ability to measure CO). WilfireSat (Johnston et al., 2020) will help with the FRP and hotspot count in the afternoon during the peak of the fire activity.

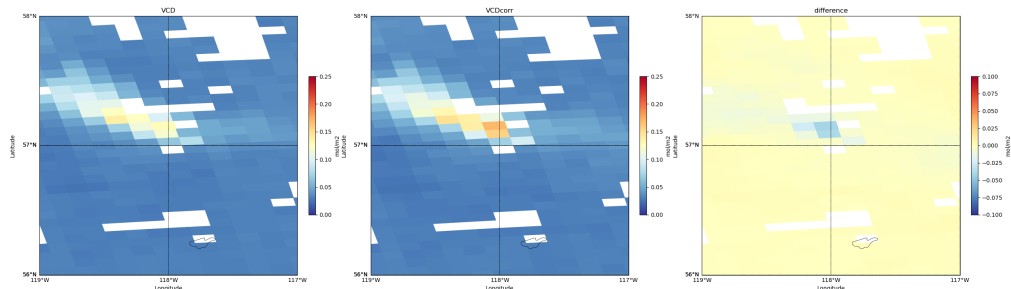

**Figure B1.** Effect of applying the TROPOMI averaging kernel for a fire that overlapped well with the GEM-MACH model (using GF-FEPS fire emissions). The averaging kernel corrected columns are estimated using the corrected averaging kernel $AVKcorr = \int_0^z AVKz \cdot Nz\,dz / \int_0^z Nz\,dz$ that is then applied to the CO colummns: VCDcorr = VCD/AVKcorr, where Nz is the GEM-MACH CO profile. The original CO columns are shown on the left panel, the corrected ones in the middle panel, and the difference (VCD-VCDcoor) on the right panel. The emissions were estimated with the original and corrected VCDs and show an increase of 17 % when the corrected VCDs are used. The example shown is for the same fires as in Fig. 5 top panel (May 21, 2019 at 57°N and 118°).

## Appendix B:  Effect of the TROPOMI averaging kernel

## Appendix C:  Regions and Biomes

### C1    GLC2000

## Appendix D:  TROPOMI GFFEPS comparison

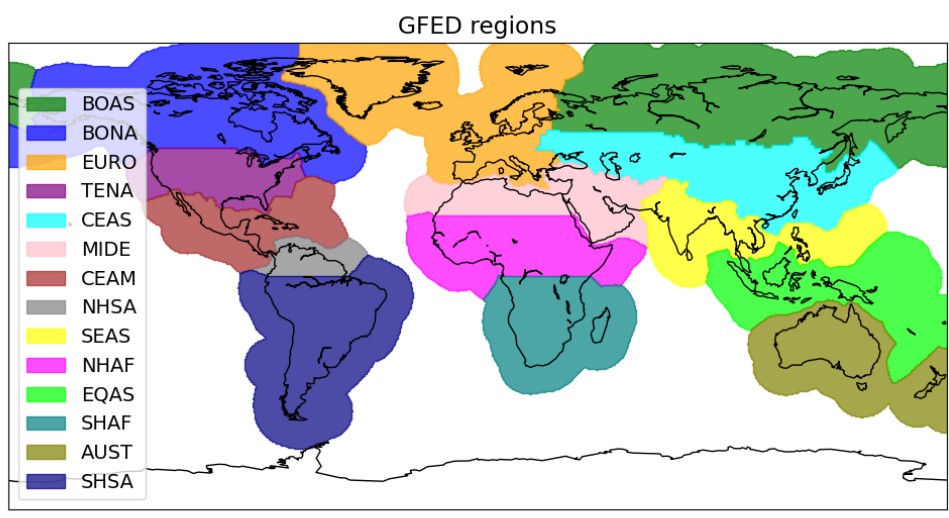

**Figure C1.** Polygons used to define the geographical region, these are based on the GFED regions from Giglio et al. (2003). Abbreviations are as follows: boreal North America (BONA), temperate North America (TENA), Central America (CEAM), Northern Hemisphere South America (NHSA), Southern Hemisphere South America (SHSA), Europe (EURO), Middle East (MIDE), Northern Hemisphere Africa (NHAF), Southern Hemisphere Africa (SHAF), boreal Asia (BOAS), Central Asia (CEAS), Southeast Asia (SEAS), equatorial Asia (EQAS), and Australia and New Zealand (AUST).

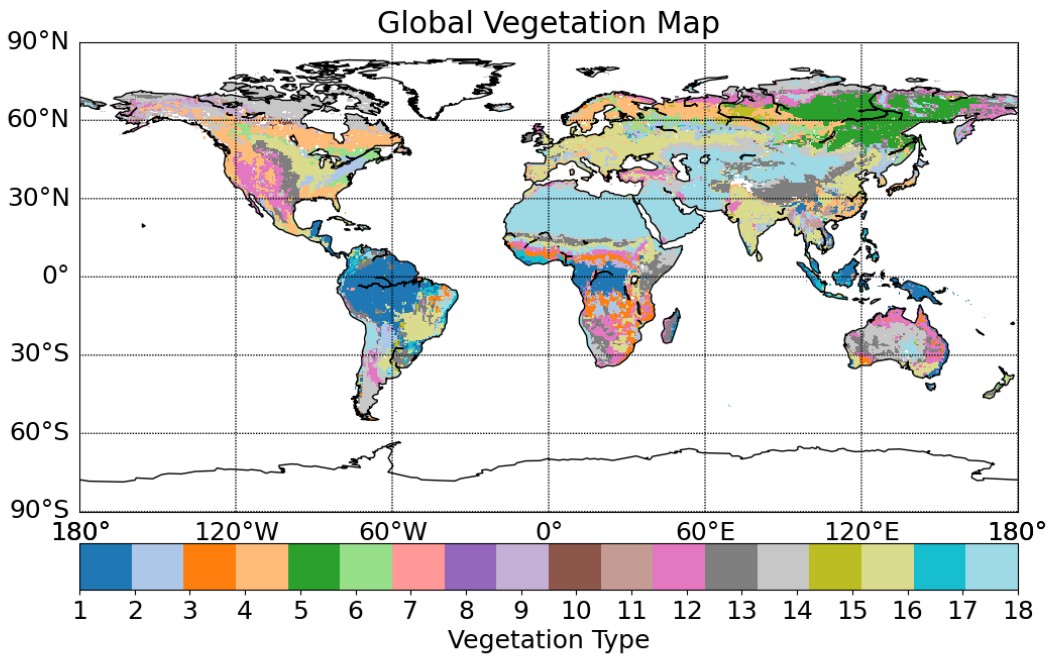

**Figure C2.** GLC2000 biomes, see Bartholomé and Belward (2005). The definition 1 to 18 can be found in the table below (Table C1).

**Table C1.** GLC2000 biomes, see Bartholomé and Belward (2005), the extend of the biomes is also included in the table in $km^2$ as well as in percentage relative to the total Earth surface (assuming 510 million $km^2$). Note that the GLC2000 dataset does not cover the entire globe, it covers the areas between 90°N and 56°S, therefore the area of water and snow and ice are only taking anything north of 56°S into account.

| Number | Description | area ($km^2$) | area (%) |
|:---:|:---|:---:|:---:|
| 1 | Tree Cover, broadleaved, evergreen LCCS >15 | $1.2e^7$ | 2.4 |
| 2 | Tree Cover, broadleaved, deciduous, closed | $6.5e^6$ | 1.3 |
| 3 | Tree Cover, broadleaved, deciduous, open | $3.7e^6$ | 0.7 |
| 4 | Tree Cover, needle-leaved, evergreen | $9.1e^6$ | 1.8 |
| 5 | Tree Cover, needle-leaved, deciduous | $3.8e^6$ | 0.7 |
| 6 | Tree Cover, mixed leaf type | $3.2e^6$ | 0.6 |
| 7 | Tree Cover, regularly flooded, fresh water | $5.7e^5$ | 0.1 |
| 8 | Tree Cover, regularly flooded, saline water | $1.1e^5$ | 0.02 |
| 9 | Mosaic: Tree cover / Other natural vegetation | $2.4e^6$ | 0.5 |
| 10 | Tree Cover, burnt | $3.0e^5$ | 0.05 |
| 11 | Shrub Cover, closed-open, evergreen | $2.1e^6$ | 0.4 |
| 12 | Shrub Cover, closed-open, deciduous | $1.1e^7$ | 2.2 |
| 13 | Herbaceous Cover, closed-open | $1.3e^7$ | 2.6 |
| 14 | Sparse Herbaceous or sparse Shrub Cover | $1.3e^7$ | 2.7 |
| 15 | Regularly flooded Shrub and/or Herbaceous Cover | $1.7e^6$ | 0.3 |
| 16 | Cultivated and managed areas | $1.7e^7$ | 3.4 |
| 17 | Mosaic: Cropland / Tree Cover / Other natural vegetation | $3.5e^6$ | 0.7 |
| 18 | Mosaic: Cropland / Shrub or Grass Cover | $3.1e^6$ | 0.6 |
| 19 | Bare Areas | $2.0e^7$ | 3.9 |
| 20 | Water Bodies (natural & artificial) | $3.3e^8$ | 65.4* |
| 21 | Snow and Ice (natural & artificial) | $2.8e^5$ | 0.6** |
| 22 | Artificial surfaces and associated areas | $2.8e^5$ | 0.05 |

*Does not include water south of 56°S. ** Does not include Antarctica.

**Table D1.** TROPOMI-derived CO emissions versus GFFEPS CO emissions for different regions around the globe.

| Region | Slope | R | RMSE | sample size |
|--------|-------|------|------|-------------|
| BONA | 0.31 | 0.36 | 1976 | 183 |
| TENA | 0.42 | 0.51 | 1074 | 45 |
| CEAM | 1.80 | 0.59 | 401 | 22 |
| NHSA | 0.82 | 0.74 | 249 | 58 |
| SHSA | 0.42 | 0.33 | 844 | 789 |
| EURO | 0.20 | 0.44 | 215 | 7 |
| MIDE | -0.60 | -0.90 | 300 | 3 |
| NHAF | 0.19 | 0.31 | 446 | 557 |
| SHAF | 0.14 | 0.31 | 441 | 918 |
| BOAS | 0.35 | 0.56 | 895 | 403 |
| CEAS | 0.22 | 0.29 | 395 | 121 |
| SEAS | 0.33 | 0.07 | 579 | 203 |
| EQAS | 0.49 | 0.40 | 928 | 58 |
| AUST | 0.10 | 0.09 | 2986 | 672 |

**Table D2.** TROPOMI-derived CO emissions versus GFFEPS CO emissions for GLC2000 biomes (see Table C1 around the globe).

| Fuel Type | Slope | R | RMSE | sample size |
|-----------|-------|------|------|-------------|
| 1 | 0.14 | 0.02 | 2794 | 706 |
| 2 | 0.14 | 0.23 | 866 | 356 |
| 3 | 0.25 | 0.41 | 885 | 768 |
| 4 | 0.47 | 0.44 | 1227 | 233 |
| 5 | 0.34 | 0.54 | 1008 | 248 |
| 6 | 0.32 | 0.77 | 1988 | 36 |
| 9 | 0.25 | 0.59 | 633 | 78 |
| 11 | 0.11 | 0.07 | 1545 | 150 |
| 12 | 0.40 | 0.53 | 444 | 519 |
| 13 | 0.25 | 0.34 | 640 | 193 |
| 14 | 0.61 | 0.35 | 395 | 172 |
| 15 | 0.25 | 0.37 | 564 | 85 |
| 16 | 0.46 | 0.34 | 199 | 131 |
| 17 | 0.32 | 0.40 | 806 | 166 |
| 18 | 0.12 | 0.73 | 1342 | 107 |

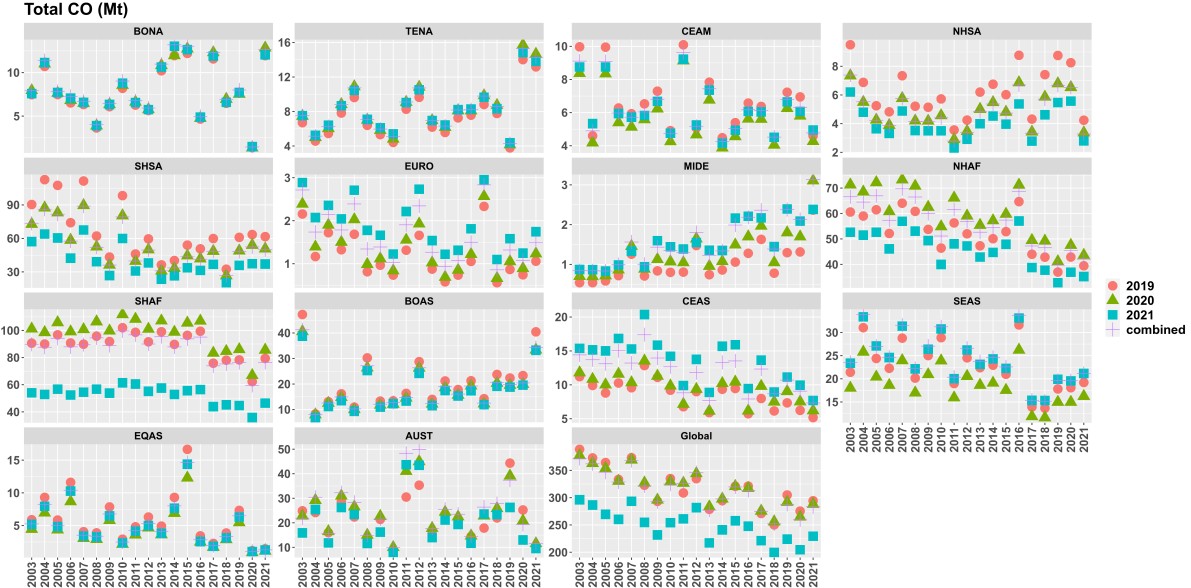

**Figure E1.** TROPOMI-FRE emissions between 2003 and 2021 based on different emission coefficients estimated from single years (2019-2021, see Tables E1, E2, and E3), and combined years (2019-2021, see Table 2). While individual $EC_{CO}$ are changing overall the change is small particularly in using TROPOMI emission estimates from 2019, 2020 and "combined" (2019-2021). The $EC_{CO}$ in 2021 is significantly lower for biomes 1-3, leading to lower emission estimates of approximately 60 Mt (CO) globally ($\sim 20\%$).

## Appendix E: $EC_{CO}$ for different years

## Appendix F: Emission factors

**Table E1.** Same as Table 2, but for 2019 only. Note 2019 was the year of the "black summer" fires in Australia that burned mostly type 1 vegetation, those were extreme fires with very thick smoke.

| Number | Description | $EC_{CO}$ (g/MJ) | R | sample size | Rank |
|---|---|---|---|---|---|
| 1 | Tree Cover, broadleaved, evergreen | 150 | 0.55 | 498 | 1 |
| 2 | Tree Cover, broadleaved, deciduous, closed | 100 | 0.61 | 194 | 4 |
| 3 | Tree Cover, broadleaved, deciduous, open | 53 | 0.71 | 357 | 2 |
| 4 | Tree Cover, needle-leaved, evergreen | 56 | 0.47 | 183 | 8 |
| 5 | Tree Cover, needle-leaved, deciduous | 111 | 0.47 | 220 | 10 |
| 6 | Tree Cover, mixed leaf type | 67 | 0.65 | 31 | 15 |
| 9 | Mosaic: Tree cover / Other natural vegetation | 72 | 0.82 | 44 | 12 |
| 11 | Shrub Cover, closed-open, evergreen | 100 | 0.78 | 89 | 13 |
| 12 | Shrub Cover, closed-open, deciduous | 28 | 0.37 | 220 | 3 |
| 13 | Herbaceous Cover, closed-open | 58 | 0.5 | 82 | 6 |
| 14 | Sparse Herbaceous or sparse Shrub Cover | 31 | 0.28 | 103 | 9 |
| 15 | Regularly flooded Shrub and/or Herbaceous Cover | 67 | 0.54 | 64 | 14 |
| 16 | Cultivated and managed areas | 28 | 0.36 | 62 | 5 |
| 17 | Mosaic: Cropland / Tree Cover / Other natural vegetation | 58 | 0.43 | 83 | 7 |
| 18 | Mosaic: Cropland / Shrub or Grass Cover | 69 | 0.46 | 60 | 11 |

**Table E2.** Same as Table 2, but for 2020 only.

| Number | Description | EC$_{CO}$ (g/MJ) | R | sample size | Rank |
|---|---|---|---|---|---|
| 1 | Tree Cover, broadleaved, evergreen | 101 | 0.71 | 459 | 1 |
| 2 | Tree Cover, broadleaved, deciduous, closed | 93 | 0.57 | 225 | 4 |
| 3 | Tree Cover, broadleaved, deciduous, open | 79 | 0.42 | 346 | 2 |
| 4 | Tree Cover, needle-leaved, evergreen | 61 | 0.52 | 151 | 8 |
| 5 | Tree Cover, needle-leaved, deciduous | 90 | 0.43 | 294 | 10 |
| 6 | Tree Cover, mixed leaf type | 47 | 0.28 | 5 | 15 |
| 9 | Mosaic: Tree cover / Other natural vegetation | 67 | 0.22 | 47 | 12 |
| 11 | Shrub Cover, closed-open, evergreen | 75 | 0.26 | 103 | 13 |
| 12 | Shrub Cover, closed-open, deciduous | 43 | 0.76 | 227 | 3 |
| 13 | Herbaceous Cover, closed-open | 45 | 0.30 | 139 | 6 |
| 14 | Sparse Herbaceous or sparse Shrub Cover | 56 | 0.38 | 30 | 9 |
| 15 | Regularly flooded Shrub and/or Herbaceous Cover | 124 | 0.63 | 89 | 14 |
| 16 | Cultivated and managed areas | 38 | 0.67 | 146 | 5 |
| 17 | Mosaic: Cropland / Tree Cover / Other natural vegetation | 46 | 0.28 | 58 | 7 |
| 18 | Mosaic: Cropland / Shrub or Grass Cover | 50 | 0.34 | 56 | 11 |

**Table E3.** Same as Table 2, but for 2021 only.

| Number | Description | EC$_{CO}$ (g/MJ) | R | sample size | Rank |
|---|---|---|---|---|---|
| 1 | Tree Cover, broadleaved, evergreen | 59 | 0.52 | 507 | 1 |
| 2 | Tree Cover, broadleaved, deciduous, closed | 42 | 0.30 | 252 | 4 |
| 3 | Tree Cover, broadleaved, deciduous, open | 28 | 0.22 | 439 | 2 |
| 4 | Tree Cover, needle-leaved, evergreen | 55 | 0.49 | 340 | 8 |
| 5 | Tree Cover, needle-leaved, deciduous | 92 | 0.30 | 915 | 10 |
| 6 | Tree Cover, mixed leaf type | 45 | 0.29 | 25 | 15 |
| 9 | Mosaic: Tree cover / Other natural vegetation | 65 | 0.69 | 75 | 12 |
| 11 | Shrub Cover, closed-open, evergreen | 123 | 0.53 | 101 | 13 |
| 12 | Shrub Cover, closed-open, deciduous | 34 | 0.43 | 295 | 3 |
| 13 | Herbaceous Cover, closed-open | 36 | 0.50 | 173 | 6 |
| 14 | Sparse Herbaceous or sparse Shrub Cover | 78 | 0.52 | 49 | 9 |
| 15 | Regularly flooded Shrub and/or Herbaceous Cover | 52 | 0.49 | 56 | 14 |
| 16 | Cultivated and managed areas | 94 | 0.60 | 130 | 5 |
| 17 | Mosaic: Cropland / Tree Cover / Other natural vegetation | 85 | 0.34 | 68 | 7 |
| 18 | Mosaic: Cropland / Shrub or Grass Cover | 63 | 0.88 | 87 | 11 |

**Table F1.** Emissions factors for CO from this work using a variable conversion factor "variable" from (Kaiser et al., 2012) and a universal conversion factor "universal" of 0.368kg/MJ (Wooster et al., 2005) to the emission coefficients as in Table 4. The emission factors (from this study) depend greatly on the conversion factor which differs by as much as a factor of four. The emission factors for GFED and GFAS are listed for comparison.

| Description | conversion factor (kg/MJ) | variable(g/kg) | universal (g/kg) | GFED (g/kg) | GFAS (g/kg) |
|---|---|---|---|---|---|
| AGRI | 0.29 | 169 | 133 | 102 | 92 |
| BORF | 1.55 | 59 | 247 | 127 | 106 |
| DEFO | 0.96 | 54 | 141 | 93 | NA |
| PEAT | 5.87 | 31 | 497 | 210 | 210 |
| SAVA | 0.78 | 77 | 163 | 63 | 61 |
| TEMF | 0.49 | 224 | 299 | 88 | 101 |

*Data availability.* TROPOMI data can be downloaded from https://s5phub.copernicus.eu. The MODIS fire product is publicly available for download at: http://modis-fire.umd.edu/index.php. The location of fires and the TROPOMI CO emission estimates can be found here: https://collaboration.cmc.ec.gc.ca/cmc/arqi/Griffin_et_al_fireco/

*Author contributions.* DG, CAML, and ED developed the emissions estimation methods. KA, JC, and PM developed GFFEPS and prepared the GEM-MACH model run. AK and AF contributed to the data visualization. DG carried out the data analysis and prepared the article with contributions from all co-authors.

*Competing interests.* The authors declare no competing interests.

*Acknowledgements.* This work contains modified Copernicus Sentinel data. The Sentinel 5 Precursor TROPOMI Level 2 product is developed with funding from the Netherlands Space Office (NSO) and processed with funding from the European Space Agency (ESA). The MODIS data set was provided by LANCE FIRMS operated by NASA ESDIS with funding provided by NASA Headquarters. This study contains modified Copernicus Atmosphere Monitoring Service Information [2019]; neither the European Commission nor ECMWF is responsible for any use that may be made of the information it contains.

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
