# Peer review of "Biomass Burning CO Emissions: Exploring Insights through TROPOMI-derived Emissions and Emission Coefficients"

_EGUsphere, 2023_

## Author Comment (AC1)

We extend our gratitude to Reviewer 2 for their detailed review. In response to the feedback from both reviewers, we have implemented significant revisions to the manuscript. We offer detailed insights into these changes below, and for your convenience, we have also provided a PDF document illustrating the differences between the previous and current versions.

Our answers are highlighted in red font, while the reviewers' comments are in black.

This manuscript focuses on carbon monoxide (CO) emissions from wildfires; its goal is to produce a ~20 year global budget based on emission coefficients derived from TROPOMI and MODIS data:

- First, TROPOMI-derived emissions are calculated for the 2019-2021 period with an automated version of the flux method. To verify the robustness and uncertainties of the automated method, emissions are derived from synthetic CO column values generated with the GEM-MACH model and CFFEPS for the May-September 2019 period over North America. From this test, several filters (dealing with background CO, wind, plume geometry, etc.) are defined.

- Then, global TROPOMI-derived emissions are calculated for the 2019-2021 period, the filters defined in the previous step are applied, and CO emission coefficient values (EC) are calculated separately for 15 biomes: EC=E/FRP=emissions/fire radiative power. FRP values are from MODIS Aqua. Somehow

the GFAS FRP dataset (which results from assimilating both MODIS Terra and Aqua FRP values in GFAS) is used to provide "a guidance on total daily FRP that can then be combined with the derived ratio between TROPOMI CO emissions and MODIS FRP"; meaning unclear from the manuscript.

To make it clearer, we added:

"… by applying the derived emission coefficients to assimilated daily FRP based on MODIS measurements (available from GFAS)."

- Following, a global budget of CO emissions from fires is calculated for the 2003-2021 period based on TROPOMI-MODIS Aqua EC values and GFAS FRP values. Results are analyzed by region and by biome, and compared with respect to several emission inventories.

The manuscript should be improved to avoid repetitions, typos, correct wording and sentence integrity, provide explanations for acronyms/abbreviations when first used, etc. The text is hard to follow at times because of these issues. Some of the Appendices are never mentioned in the text. Expressions such as "many of the retrieved emissions", "many outliers", "aligns very well", "the model agrees pretty well" should be avoided. Quantitative statements should be used instead.

We reviewed the manuscript, revising numerous sections to enhance its quality by eliminating repetitions, rectifying typographical errors, and refining the wording. Details can be found below, a pdf showing the difference between the last and current version is provided.

We specifically changed the expressions high-lighted by the reviewer:

"*Many of the retrieved* emissions are very close to the original emissions, however, *many outliers* can be seen, where most likely the retrieved values are below the originals." -> "While a substantial portion of the retrieved emissions closely matches the original values, there are noticeable outliers, where the retrieved values are below the original emissions."

"This shows that for certain regions (i.e. CEAM, NHSA, EURO, and MIDE) the *model agrees pretty well* with the satellite-derive emissions (taking slope, R and RMSE into consideration)." -> "Considering factors like slope, R (correlation coefficient), and RMSE (root mean square error), the model demonstrates strong agreement with satellite-derived emissions for specific regions, namely CEAM, NHSA, EURO, and MIDE (see Table B1)."

Some figures would benefit from more consistent axis ranges (e.g., Fig. 1 b and c), axis labels with both axis title and units (Fig. 5). Panels labels should be properly referred to in text (e.g., Fig. 5). For clarity, please add a map showing all the regions discussed in the manuscript (CEAM, NHSA, EURO, ...). Consider including a figure illustrating the areal extent of biomes.

Fig1c: axis range has been adjusted to be similar to 1band a grid included.

Fig5: missing axis labels and units have been added.

Figures have been included showing the regions and biomes (Figs. A1 and A2), and refer to these within the manuscript.

The manuscript should justify why the TROPOMI averaging kernels are not applied to retrievals with qa_value <1. From the TROPOMI CO readme document: "We recommend using only data with a qa_value = 1 in case the averaging kernel is not applied. Data with a qa_value = 0.7 are of similar quality provided the averaging kernel is used to account for the vertical retrieval sensitivity in the presence of mid-level clouds. Quality assurance values of qa_value = 0.4 represent experimental data to be used with caution." Table 3 in that same document provides additional information regarding qa_values, cloud heights, and tau_aer values.

While the masking effect of smoke and clouds on FRP observations is discussed several times, the effect of smoke and clouds on TROPOMI retrievals is never mentioned. This is a very important issue, given the focus on TROPOMI observations acquired over active fires, with very smoky and (potentially) very cloudy conditions. Please discuss.

While for clear sky the TROPOMI CO averaging is around 1, the averaging kernel is affected by clouds. Near fires there is mostly smoke, which is treated as clouds in the TROPOMI retrieval algorithm. Smoke is made of small particles (like 0.1 um) and so should be all but invisible to light at 2 um. Cloud particles are much larger and so you cannot use the cloud Aks to estimate the AK with smoke, thus it would not be helpful to correct for the averaging kernel in the case of fire emissions as those might be wrong to begin.

Schneising et al., 2020 (https://doi.org/10.5194/acp-20-3317-2020) found the TROPOMI averaging kernel to be 0.95±0.05 in the boundary layer, and thus at the most will only account for a 5% uncertainty which is much lower than the overall uncertainty of the emissions estimate.

We also found a recent publication by Rowe et al., 2022 (https://pubs.acs.org/doi/full/10.1021/acsearthspacechem.2c00048) that investigates the uncertainty of the TROPOMI CO observations in smoke, we added the reference to our manuscript, showing approximately 10% higher CO than the aircraft measurements. This uncertainty is consisted with the uncertainty assumed in our error budget analysis (Table 1).

We included the following discussion in the manuscript: "For our analysis, we have utilized observations rated with a quality flag greater than 0.5, where 0 represents the lowest quality and 1 denotes the highest quality. This choice aligns with the recommended quality threshold (Apituley et al., 2018). Notably, when we investigate areas near active fires, the quality flag of the retrieval can be impacted by the presence of smoke. Consequently, including observations with a quality flag of 1 would result in the exclusion of a substantial number of data points, primarily due to the influence of smoke.
The CO averaging kernel from the TROPOMI observations predominantly registers

values close to 1 within the boundary layer, specifically around 0.95 with a narrow range of variability (approximately ±0.05) (Schneising et al., 2020). Nevertheless, the presence of clouds diminishes the sensitivity of the averaging kernel beneath them. It is important to note that smoke primarily comprises minuscule particles (~1 μm), which are invisible at the 2μm wavelength. The TROPOMI algorithm lacks the capability to differentiate between clouds and smoke, which is why attempting to correct the averaging kernel in regions near fires would introduce additional uncertainty to the analysis. Thus, the averaging kernel is not considered in this study, but is taken into account in the overall uncertainty (uncertainty of VCDs) of the emission estimate (see Table 1). Rowe et al. (2022) investigated TROPOMI CO in thick fire plumes and found agreement within 10 % with the aircraft observations, which has been used to estimate the overall uncertainty of the emissions."

In lines 319-321, to explain discrepancies between emissions from TROPOMI and GFFEPS, it is suggested that GFFEPS values may be low because of missed fires due to thick smoke. Could TROPOMI values be high due to smoke? Clouds? Other factors?

 GFFEPS relies on hotspot detection for the emission estimation (not missed fires, missed hotspots), as shown in Fig. 5 (bottom panel), some of the hotspots are under a layer of thick smoke meaning these are not accounted for in GFFEPS, reducing the emissions.  The figure also shows that the TROPOMI observations are high over the fire and the smoke plume, and TROPOMI provides good quality data that meet the quality threshold. If anything, the thick clouds would lead to lower CO VCDs (not higher) as the averaging kernel would have lower sensitivity in the boundary layer below the cloud. The TROPOMI CO emissions approach has been validated for fires in North America using measurements from FIREX-AQ (Stockwell et al., 2022), the uncertainties associated with the emission direct estimate are discussed in section 2.3, showing an overall uncertainty of 40%. GFFEPS on the other hand is a new fire emissions prediction system and has not been validated, and neither does it have a range of uncertainties. We summarized the previous discussion and included this in the manuscript:

"This discrepancy is likely due to an underestimation from GFFEPS rather than an overestimation of TROPOMI emissions: TROPOMI offers high-quality data over fires and smoke plumes (see Fig 5) and potential cloud cover would likely result in lower CO levels detected by TROPOMI due to reduced sensitivity below the smoke plume rather than higher. It is important to note that the TROPOMI CO emissions approach has been validated (Stockwell et al., 2022) with a 40% overall uncertainty see Sect. 2.3), while GFFEPS still requires validation and associated uncertainty estimates."

Figure 6 shows that the correlation between CO and FRP may not be very robust, despite R being 0.70. How would the slope (i.e., EC) change if the most extreme outliers were removed one by one?

We removed the highest values (although it's hard to say that these are outliers, these might just be large fires for which the emission retrieval is often not successful or they might not be as common). The result is shown below. While R reduces as more values are removed by 0.12, it is still above 0.5. The slope increases when the highest values are removed, however, when all 4 values are removed the slope is back to the original slope shown in the paper. The increase of the slope is at maximum 16% and below the estimated uncertainty and range of values for other years.

[Figure]

Furthermore, we included the analysis for separate years to analyse the impact: while 2019 and 2020 are not too different, 2021 shows smaller emissions, primarily and most significantly for the SHAF region. The global totals are almost identical for 2019, 2020, and 2019-2021. The following figure is now included in the appendix and can be interpreted as an indicator of uncertainty for the total emission estimates.

[Figure]

Line 379: "The emission coefficients vary between 120 and 39 g/MJ" Please clarify that EC=120 value corresponds to (11): shrub cover, closed-open, evergreen; EC=39 to (12): shrub cover, closed-open, deciduous. The most extreme EC values correspond to rather similar biomes, please discuss.

 The error has been corrected: "The emission coefficients vary between 120 and 39 g/MJ, where the largest CO emissions relative to FRP are from shrub cover evergreen (11) and the lowest are from shrub cover deciduous (12)."

Even though both types are shrub, they are very different types. Type 11 is not very common (~0.5% of the Earth surface area) and predominantly occurs in central Asia, in some parts of Northern Canada and Alaska. Deciduous shrub is more common (~2.2%) and spread around the entire globe. This is showing that just using shrub as a vegetation classification might not be a good idea when estimating fire emissions. This can be seen in the figure below (only vegetation 11 and 12 are shown – otherwise it's the same figure as Fig. A2)

We included the following in the manuscript:

"Even though both biomes are shrub, they are quite different biomes, based on their CO emissions and way hey burn as well their location and occurrence. Evergreen shrub (biome 11) is not very common (it covers approximately 0.5\,\% of the Earth's surface) and appears primarily in Central Asia and in some parts of Northern Canada and Alaska (see Fig. A2 and Table A1). Whereas deciduous shrub covers approximately 2.2% of the Earth's surface and grows globally (see Fig. A2 and Table A1). ....  A simplified

classification of forest, shrub and grassland is not appropriate based on our results. For example the ECco for different types of shrubs has both the largest and smallest emission coefficients and forests vary between roughly 49 and 95 g/MJ."

[Figure]

Lines 379-380: "where the largest CO emissions relative to FRP are from broadleaved evergreen tree cover (1) and the lowest are from cultivated managed areas (16)". How could readers see CO emissions relative to FRP? Similar issue in lines 474-475.

 We included the following in the introduction where the term appears for the first time:

"Additionally, we determine biome specific emission coefficients (emissions relative to FRP), which are the CO emissions produced relative to the amount of heat energy released by the fire (FRP). This can provide insights into the efficiency of combustion, and help quantifying how emissions from a particular ecological region or biome are related to the heat energy generated by wildfires in that region. This information can be valuable for understanding the environmental impact of wildfires in different ecosystems and for developing strategies to manage and mitigate their effects. Furthermore FRP is often and more easily measured from satellites compared to CO, and determining a biome specific CO-to-FRP ratio can help to determine the daily total emissions of fires."

We added to the sentence in Sect 4 (near the former l.379-380):

"The emission coefficients vary between 120 and 39 g/MJ, where the largest CO emissions relative to FRP are from shrub cover evergreen (11) and the lowest are from shrub cover deciduous (12), meaning three times more CO is emitted from evergreen shrub (biome 11) compared to deciduous shrub (biome 12) for a fire that burns with equivalent heat energy."

Line 401: "Most CO emissions are from evergreen forests (biome type 1), which also has the largest EC for CO". Table 2 shows that biome 11 has the largest EC, please correct.

Thank you for pointing out this error, we corrected the statement to: "... which also has one of the largest EC for CO..."

Please justify why is biome 15 ("Regularly flooded shrub and/or herbaceous cover") included in the analysis. Biomes 7 and 8 are excluded because "fires were not observed, namely: regularly flooded tree cover (7 and 8)"

No fire emissions were detected for these types of biomes (7, 8, 10, 19, 20, 21, 22) by TROPOMI. For regularly flooded shrub and herbaceous cover TROPOMI observed 127 fires. We added the following statement in the manuscript: "...fires were not observed by TROPOMI (and therefore no information is available on the CO emissions) ..."

Biome # 15 (regularly flooded shrub and/or herbaceous cover; should this biome be included in the analysis, given that it is regularly flooded?) has the second largest EC (105 g/MJ). Please explain.

The biome is included because fires and CO emissions could be detected for this type of biome. With TROPOMI we were able to estimate emissions from 127 fires.

The manuscript assumes that EC values "do not change drastically over the years". However, tables C1, C2, and C3 in Appendix C (never mentioned in the manuscript; please correct) show otherwise. EC values change between -50% and +150%, depending on the period analyzed. EC values change strongly among biomes; changes do not seem to follow recognizable patterns. How would the global budget change if a different set of EC values was used? Please quantify.

We estimated emissions with 2018, 2019, 2021 values, and included the comparison in the appendix (Fig. C1). Even though the EC change over the years the impact in the total

emissions is not as significant. The largest difference appears for 2021 where the emission coefficients particularly for biomes 1-3 is about half as much as for the other years. This reflects to a 20% change in the emission totals.

We included a brief introduction in Sect. 4:

"The slope between the CO emissions (in g/s) and the FRP (in MJ/s) is EC (in g/MJ), the values of which are shown in Table 1 (and for 2019, 2020, and 2021 in Tables C1, C2, and C3, respectively)."

And added a discussion in Sect. 5.1:

"To assess the uncertainty of the total annual emissions of our estimates (TROPOMI-FRE), we also used emission coefficients derived from fires of individual years (2019 to 2021). Using emission coefficients from 2019,2020, and 2019-2021 combined did not impact the total emissions (see Fig. C1), only for 2021 the total emissions reduced by approximately 20%, due to overall lower ECco (for biomes 1-3, see Table C3). This shows that the uncertainty of our approach is at least 20%, but since the individual TROPOMI derived CO emissions have an uncertainty of 40%, we would expect the overall TROPOMI-FRE annual emission to have similar uncertainties on the order of 40%."

[Figure]

What is the areal extent of each biome? That information could help readers understand what biomes (and what biome's EC) may have a stronger effect on the global budget.

We included this information in Table A1. In units of km2 as well as coverage of the Earth's surface in percent.

Figure 9 shows emissions decreasing with time. An evaluation of the temporal trend in the global budget with respect to actual measurements (not inventories) is missing in the manuscript. Have wildfire emissions really decreased in the last two decades? While measurements of global emissions for all biomes may be unavailable, a literature search may provide useful information for specific regions/biomes.

All inventories presented rely on satellite observations: GFAS (and ultimately TROPOMI-FRE) are based on satellite observations of fire hotspots (MODIS). GFFEPS is based on satellite hotspot observations, and (satellite-observed) area burned. FINN 1.5 is based on MODIS and FINN2.5 is based on MODIS/VIIRS observations. GFED also include satellite observed area burned.

The MOPITT satellite has been observing CO since 2003 Satellites alone cannot obtain total CO fire emissions, looking at the total global CO concentration or vertical column density would not distinguish the source: anthropogenic vs wildfire. MOPITT observations of CO show decrease between 2002-2018 which is consistent with our findings (Buchholz et al., 2021).

We re-wrote the entire section 5.1 "CO emissions over the past two decade". We changed Figure 9 to include all inventories discussed in this paper (with the exception of GFFEPS that is currently only available for 2019). We included a Figure, showing the change of emissions for different regions from all inventories discussed in this paper (GFAS, GFED, FINN 1.5, FINN2.5). Additionally, we provided a table with the trends and indicate whether or not the trend is significant (for all 5 inventories, including our own).

We included a literature search finding that area burned and MOPITT CO are declining [Buchholz et al., 2021, Zheng et al., 2021, Giglio et al., 2013], which is consistent with our findings. However, we did not find a publication focussing fire emission trends for the same time period as our study.

Section 5.1:

"5.1 CO emissions over the past two decades

The inventories discussed in the previous section provide data for various past years, except for GFFEPS (currently only available for 2019). For our independent estimates, we relied on daily FRP data from GFAS, which is based on MODIS FRP, available from 2003 to the present. Under the assumption that the ECco values (as derived in Sect. 4) remain relatively stable over the years, we conducted an extensive analysis of the

entire time series and calculated CO emissions spanning from 2003 to 2021 (refer to Fig. 10). To assess the uncertainty of the total annual emissions of our estimates (TROPOMI-FRE), we also used emission coefficients derived from fires of individual years (2019 to 2021). Using emission coefficients from 2019,2020, and 2019-2021 combined did not impact the total emissions (see Fig. C1), only for 2021 the total emissions reduced by approximately 20 %, due to overall lower ECCO (for biomes 1-3, see Table C3). The uncertainty of our approach is at least 20 %, but since the individual TROPOMI derived CO emissions have an uncertainty of 40%, we would expect the overall TROPOMI-FRE annual emission to have similar uncertainties on the order of 40 %. Furthermore, we also present data from the other four inventories for the same time frame. The results are visualized in Fig. 9. As expected, the emissions from wildfires in various regions across the globe exhibit significant interannual variability.

Notably, EURO and MIDE consistently report the lowest wildfire emissions throughout the entire time series and are barely noticeable in the figures. The predominant source of wildfire CO emissions is from SHAF and SHSA, followed by NHAF. This consistent pattern is evident for all the inventories analyzed.

To enhance the clarity of emissions identification and changes across different regions, we have depicted emissions by region in Fig. 10. The rate of change for this time period has been quantified for each inventory, and the results are presented in Table 3. Significant rates of change (with a p-value below 5 %) are highlighted in bold, while all other rates of change are statistically insignificant.

Globally, CO emissions are experiencing a decrease ranging from 5.1 to 8.7 Mt(CO)/yr between 2003 and 2021 across all inventories, with the exception of GFED. Notably, GFED does not reflect a global decrease due to the substantial increase in CO emissions within the BOAS region, amounting to 19.8 Mt(CO)/yr. This overall decrease is primarily driven by significant reductions in SHSA (ranging from 2.1 to 6.3 Mt(CO)/yr), NHAF (ranging from 0.6 to 7.6 Mt(CO)/yr), SHAF (ranging from 0.9 to 5.6 Mt(CO)/yr), and CEAS (ranging from 0.3 to 3.3 Mt(CO)/yr), all of which show statistically significant decreases across at least four inventories. In contrast, CO emissions from wildfires are on the rise in TENA, with an increase ranging from 0.2 to 4.1 Mt(CO)/yr. Additionally, emissions in the EQAS region exhibit an interannual cycle that appears to correlate with El Niño years, resulting in higher emissions across all inventories in 2006, 2009, 2014, 2015, and 2019.

These findings align with prior research. Giglio et al. (2013) reported a decreasing trend in the annual area of land burned since 2000, which corroborates our observed reduction in CO emissions. Moreover, Zheng et al. (2021), also observed a decline in burned area between 1998 and 2015 through satellite observations, but reported

stable or only slight decreases in wildfire emissions. The satellite instrument "Measurement of Pollution in the Troposphere" (MOPITT) on board the TERRA satellite (Drummond et al., 2010) has been observing CO since 2000, (Buchholz et al., 2021) showed that MOPITT CO has been steadily decreasing by -0.50% per year between 2002 to 2018. No study examining fire emissions for the time period presented here currently exists to our knowledge."

Furthermore, we included some changes in the conclusion section to reflect the changes in section 5.1.

"Examining the trends over the past two decades (corresponding to the MODIS lifetime), it appears that global CO wildfire emissions have, on the whole, decreased. This decline is consistently observed across all inventories utilized in this study. However, this trend is highly region-specific, with the most substantial reductions occurring in SHSA, SHAF, NHAF, and CEAS. Conversely, wildfire emissions in TENA are on the rise. For all other regions, the variability within the past two decades has been too substantial to determine a statistically significant trend."

References:

Buchholz, R. R., Worden, H. M., Park, M., Francis, G., Deeter, M. N., Edwards, D. P., Emmons, L. K., Gaubert, B., Gille, J., Martínez-Alonso, S., Tang, W., Kumar, R., Drummond, J. R., Clerbaux, C., George, M., Coheur, P.-F., Hurtmans, D., Bowman, K. W., Luo, M., Payne, V. H., Worden, J. R., Chin, M., Levy, R. C., Warner, J., Wei, Z., and Kulawik, S. S.: Air pollution trends measured from Terra: CO and AOD over industrial, fire-prone, and background regions, Remote Sensing of Environment, 256, 112, https://doi.org/https://doi.org/10.1016/j.rse.2020.112275, 2021.

Giglio, L., Randerson, J. T., and van der Werf, G. R.: Analysis of daily, monthly, and annual burned area using the fourth-generation global fire emissions database (GFED4), Journal of Geophysical Research: Biogeosciences, 118, 317–328, https://doi.org/https://doi.org/10.1002/jgrg.20042, 2013.

Zheng, B., Ciais, P., Chevallier, F., Chuvieco, E., Chen, Y., and Yang, H.: Increasing forest fire emissions despite the decline in global burned area, Science Advances, 7, eabh2646, https://doi.org/10.1126/sciadv.abh2646, 2021.

Lines 446-447: "certain regions see increased emissions (e.g. TENA, AUST)" (text refers to Fig. 9). Hard to know for sure, but it looks like neither TENA nor AUST show increasing emissions. Either provide a better figure to illustrate the statement or remove statement. Similar issue in line 498.

We re-wrote the entire section 5.1, see above. We included an additional figure that makes it easier to see the trends by regions by plotting the emissions by region

normalized to 2003. We included the slope of the line of best fit in a table in the appendix.

Additionally, to the TROPOMI/FRE dataset we also looked at GFED, GFAS and FINN between 2003 to 2021.

We changed the conclusion section based on the changes from this section.

Lines 476-477: "for forests we determined ECs between 64 and 120 g/MJ" Please note that EC=120 is not for a forest but for biome 11: Shrub Cover, closed-open, evergreen.

Thank you for pointing out our typo, we corrected this in the manuscript:

"(e.g. for forests we determined ECs between 64 and 95 g/MJ)"

The manuscript refers to the need for multiple observations in a day (from geostationary instruments) in order to better understand fire evolution; TEMPO is mentioned but GEMS (which has been operating for a couple of years now) is not. Please discuss both.

We appended the sentence as suggested:
"Geostationary satellite sensors, such as TEMPO (covering North America), Geostationary Environment Monitoring Spectrometer (GEMS), or Sentinel-4 (covering Europe and Africa) will help to validate the diurnal pattern of emissions."

---

## Author Comment (AC2)

We would like to thank Reviewer 1 for their review. In response to the feedback from both reviewers, we have implemented significant revisions to the manuscript. I believe there might be some misunderstanding by the reviewer and much of the criticisms raised by the first reviewer do not provide valid grounds for rejection. The specific points mentioned in the review are addressed below.
Our answers are highlighted in red font, while the reviewers' comments are in black.
As a general comment we found this reviewer comments overall to be somewhat vague with little to substantiate them.

The manuscript entitled "Towards an improved understanding of wildfires CO emissions: a satellite remote-sensing perspective" by Griffin et al aims to derive CO emission coefficients by correlating TROPOMI CO with MODIS FRP and produce a new global CO emission inventory using the emission coefficients and GFAS FRP. The authors first estimate CO emissions directly for forest fires in an hour near the overpass time of S5P and evaluate temporally redistributed fire emissions from a Canada emission model – CFFEPS. By correlating TROPOMI CO flux to MODIS FRP, biome-specific CO emission coefficients are derived over different numbers of fire events. The annual budget of CO emissions is estimated finally by applying the derived CO emission coefficients to GFAS FRP, and further compared with several other inventories.

Overall, the topic of this study fits the scope of ACP well and it has very meaningful goals. As S5P TROPOMI provides CO observations of global fires at the highest spatial resolution yet, it provides a good opportunity to explore CO emission coefficient for global fires, which potentially improves the estimation of biomass burning emissions. Unfortunately, I think the study fails to achieve these goals due to the seriously flawed method for deriving emission coefficients and the failure of assessing the accuracy of CO budgets. First, I acknowledge that using TROPOMI CO observations to directly estimate CO emissions from fires during a specific short period of time is sound, and direct CO estimates are valuable independent emission data for evaluating other emission estimates. Yet, both the idea and CO estimation method are not new as TROPOMI CO has been successfully applied to assess fire emissions in several recently published papers. Now, let me lay out reasons why I think the method to derive emission coefficients is flawed.

To the best of my knowledge, at the time of submission there was no other paper containing a comprehensive global dataset of TROPOMI-derived wildfire CO emissions. Just recently (at the end of August 2023) a paper was published by Goudar et al., 2023 on plume detection and CO fire emissions using the TROPOMI dataset. Our manuscript, however, provides different approach and insights into CO fire emissions and provides a global budget, which is not the same as Goudar et al., but adds to the

scientific knowledge. Our manuscript includes a comprehensive global dataset of CO fire emission estimates using TROPOMI, including over 15000 fire emission points, which are made publicly available. The reviewer did not provide references to the claimed "several recently published papers." Another relevant publication I am aware of is the paper by Stockwell et al., 2022 (which I co-authored), where we focused on a limited number of fire emissions in North America during the FIREX-AQ campaign with the aim to validate the emission estimation method. Our study introduces an improved method that automates the emission estimate by determining the plume width for each fire plume. It also includes a comprehensive uncertainty analysis. Magro et al., 2021 published CO emissions from fires in Portugal which is not global and only includes a small sample of fires.

Other papers used MOPITT observations, which are sparse and utilize a very different approach compared to our study has been used, such as 1D-box model and has to be used (Liu et al., 2005), and Silva et al., 2017 looked into regional combustion using MOPITT.

We included the following changes to our manuscript (introduction):

"It has also been used to derive fire emissions in Portugal (Magro et al., 2021).Most recently, Goudar et al. (2023) published an automated plume detection and emission estimation algorithm utilizing TROPOMI CO, in our study, an alternative approach is explored. "

References:

Goudar, M., Anema, J. C. S., Kumar, R., Borsdorff, T., and Landgraf, J.: Plume detection and emission estimate for biomass burning plumes from TROPOMI carbon monoxide observations using APE v1.1, Geoscientific Model Development, 16, 4835–4852, https://doi.org/10.5194/gmd-16-4835-2023, 2023

Magro, C., Nunes, L., Gonçalves, O. C., Neng, N. R., Nogueira, J. M. F., Rego, F. C., and Vieira, P.: Atmospheric Trends of CO and CH4 from Extreme Wildfires in Portugal Using Sentinel-5P TROPOMI Level-2 Data, Fire, 4, https://doi.org/10.3390/fire4020025, 2021

Liu, J., Drummond, J. R., Li, Q., Gille, J. C., and Ziskin, D. C.: Satellite mapping of CO emission from forest fires in Northwest America using MOPITT measurements, Remote Sensing of Environment, 95, 502–516, https://doi.org/https://doi.org/10.1016/j.rse.2005.01.009, 2005.

Silva, S. J. and Arellano, A. F.: Characterizing Regional-Scale Combustion Using Satellite Retrievals of CO, $NO_2$ and $CO_2$, Remote Sensing, 9, https://doi.org/doi:10.3390/rs9070744, 2017

Theoretically, emission coefficient (g/J), which represents the mass of emissions per Joule radiative energy emitted from fire, can be derived if continuous, accurate rate of emission and FRP (or emission mass and FRE) are known. One MODIS instrument provides daily up to two observations of fires at the same location at low-mid latitudes. If only daytime Aqua MODIS FRP is used, it only provides one observation as with TROPOMI CO observation. In the method of deriving CO emission coefficient by correlating TROPOMI CO to Aqua MODIS FRP, the underline assumptions are that emission flux based TROPOMI CO and Aqua MODIS FRP are able to represent mean CO flux and mean FRP for a given fire sample during a specific period of time (±30min or

several hours?). I would not think these simplified assumptions hold in most cases. For MODIS FRP, it has a strong dependency on MODIS scan angle. In other words, FRP value can be largely different if the instrument observes the same fire at nadir and in large scan angles. In that case, emission coefficient likely changes largely when the scan angle varies. Moreover, the observation gaps between S5P and Aqua can be up to about 60 minutes although they are thought to be in similar orbits. I think this explains the very scattering distribution of samples in Fig 6, not to mention the very pool correlations in evergreen needle leaf dominated by forest wildfires. It looks like the authors are not aware of the characteristics of MODIS FRP except for listing the incapability of detecting very small fires and cloud/smoke contamination. A scientifically sound way would be deriving coefficients based on TROPOMI CO and FRP from the new-generation geostationary satellites, which has been done in several published papers that are never mentioned as background in Introduction nor discussed in Discussion. The accuracy of CO flux also relies on wind directions and speed, which are also a concern.

The reviewer questions the assumption made in deriving the CO emission coefficient by correlating TROPOMI CO to Aqua MODIS FRP, stating that these simplified assumptions may not hold in most cases. While it is true that we assume the emission coefficient remains constant throughout the day, introducing uncertainty, this is a common practice in top-down emission inventories that use FRP, such as GFAS. The GFAS emission coefficients do not vary throughout the day. In contrast, our study presents emission coefficients for additional biomes compared to GFAS and provides a comparison of direct emission estimates with both bottom-up and top-down approaches, which, to my knowledge, has not been done before.

We found a study by Freeborn et al., 2014 discussing the MODIS FRP uncertainty. While the uncertainty can be large for single hotspots as mentioned by the reviewer. For aggregated hotspots the uncertainty is much lower. On average, depending on the size of the fire we aggregate approximately 30 fire hotspots detected by MODIS, based on Freeborn et al., 2014 (Fig. 3) these would have a 6% uncertainty, much lower compared to our CO emission estimates that come with a 42% uncertainty and the uncertainty of the slope between FRP and CO emissions. We included the following discussion on this in the manuscript (Sect. 2.4):

"Depending on the size of the fire, we aggregate on average 30 thermal anomalies, based on Freeborn et al. (2014) this is associated with a 6 % uncertainty of the FRP, much lower compared to the uncertainty of the CO emissions estimates (42 %)."

The reviewer suggests that a scientifically sound way to derive coefficients would be based on TROPOMI CO and FRP from new-generation geostationary satellites. It is

important to note that deriving coefficients based on TROPOMI CO and FRP from geostationary satellites is not feasible for global emissions and a global emission inventory. For example, we found the following study by Zhang et al., 2012, some regions, including India are not covered. Mota et al., 2018 used geostationary FRP with the focus on emissions from South Africa. In contrast to our study, neither of these two studies combined the FR with other satellite datasets. While it may be possible to get a diurnal FRP regionally, for example, in North America with GOES-R, our aim was to present a global perspective in this study. Geostationary satellites have different characteristics, such as GOES-R in North America and SEVIRI in Africa and Europe, making their combination complex and challenging. Lastly, varying FRP is one thing, but to address the reviewer's point would one not also need diurnal emissions, making this even harder in a global sense. CO is currently not measured by geostationary satellites. Despite a second thorough literature search, we were unable to locate papers that have combined direct CO emission estimates with geostationary satellite FRP.

The reviewer mentions concerns about the accuracy of CO flux, which relies on wind directions and speed. I want to emphasize that this aspect is considered in the uncertainty estimate. Furthermore, it is essential to recognize that estimating emissions from satellite observations always involves some uncertainty from the wind speed and direction, and it is impossible to eliminate this uncertainty.

References:

Freeborn, P. H., Wooster, M. J., Roy, D. P., and Cochrane, M. A.: Quantification of MODIS fire radiative power (FRP) measurement uncertainty for use in satellite-based active fire characterization and biomass burning estimation, Geophysical Research Letters, 41, 1988–1994, https://doi.org/https://doi.org/10.1002/2013GL059086, 2014

Zhang, X., Kondragunta, S., Ram, J., Schmidt, C., and Huang, H.-C.: Near-real-time global biomass burning emissions product from geostationary satellite constellation, Journal of Geophysical Research: Atmospheres, 117, https://doi.org/https://doi.org/10.1029/2012JD017459, 2012

Mota, B. and Wooster, M. J.: A new top-down approach for directly estimating biomass burning emissions and fuel consumption rates and totals from geostationary satellite fire radiative power (FRP), Remote Sensing of Environment, 206, 45–62, https://doi.org/https://doi.org/10.1016/j.rse.2017.12.016, 2018

The accuracy of the new CO inventories depends on the accuracy of the derived CO emission coefficients and that of GFAS FRE. FRE calculation requires continuous FRP observations. Diurnal FRP varies very largely from day to day even for the same fire, especially large forest wildfires, which have been reported in several JGR and RSE papers. I would not expect reliable FRE to be calculated from daily mean GFAS FRP that is averaged using merely up to four daily MODIS FRP observations, although GFAS emissions are used in ECMWF forecast models. Furthermore, simply comparing it with a few other inventories doesn't tell any information about the accuracy of the new CO inventory. There are more than 10 BB emission inventories for different purposes, and

they can differ from each other by a factor of up to 30 in individual fire events although the difference in their annual budget could be much smaller. I don't see any meaningful contributions of a new inventory to the BB community without knowing its accuracy.

The reviewer questions the reliability of calculating fire radiative power (FRP) from daily mean GFAS FRP, which is assimilated using four or more daily MODIS FRP observations (from AQUA and TERRA). While we did not publish GFAS FRP, it is a reputable dataset that has undergone quality control and, to our knowledge, is the best currently available for our study.

I believe that the reviewer has missed the main objective of the study. The aim of our paper is not solely to publish a "better" emission inventory. In their review, the first three quarters of the paper, which encompass important aspects, have been entirely neglected. These include:
- 	The global and automated fire CO emissions estimates using TROPOMI, which is a substantial dataset (currently of over 15000 fire emissions) that we hope will be valuable to the scientific community. The aim is to eventually automate it entirely and provide CO fire emissions in near-real time.
- 	A comprehensive comparison between direct estimates, top-down and bottom-up approaches, shedding light on the strengths and weaknesses of each method.
- 	A comparison of different emission inventories and the examination of their trends. This analysis provides insights into the variations between five inventories.
We have provided uncertainties to the best of our abilities for the CO emission estimate and the emission coefficients. However, it is important to note that GFAS FRE does not include uncertainties, making it impossible to estimate uncertainties for the annual emissions. Moreover, none of the emission inventories that we compare our results to provide uncertainties.
We included an uncertainty range based on the emission coefficients found in different years:
"To assess the uncertainty of the total annual emissions of our estimates (TROPOMI-FRE), we also used emission coefficients derived from fires of individual years (2019 to 2021). Using emission coefficients from 2019,2020, and 2019-2021 combined did not impact the total emissions (see Fig. C1), only for 2021 the total emissions reduced by approximately 20%, due to overall lower ECco (for biomes 1-3, see Table C3). This shows that the uncertainty of our approach is at least 20%, but since the individual TROPOMI derived CO emissions have an uncertainty of 40%, we would expect the overall TROPOMI-FRE annual emission to have similar uncertainties on the order of 40%."

Considering the substantial variability and uncertainty inherent in fire emission models and inventories, I believe that any contribution, especially one with a completely different approach, should be welcomed to the scientific community.

To highlight the aim of our paper and make its goal clearer we included the following changes in the abstract:
- "Specifically, we use the TROPOMI (Tropospheric Monitoring Instrument) high spatial-resolution satellite datasets to create an automated and global database of burning CO emissions between 2019 and 2021."
- "A comprehensive comparison between direct estimates, top-down and bottom-up approaches, provides insight into the strengths and weaknesses of each method."

And in the introduction:
- "This approach has been entirely automated and has the capability to determine CO fire emissions in quasi near real time (as soon as TROPOMI CO and MODIS FRP observations are available)."
- "These emission coefficients can provide insights into the efficiency of combustion, and help quantifying how emissions from a particular ecological region or biome are related to the heat energy generated by wildfires in that region. This information can be valuable for understanding the environmental impact of wildfires in different ecosystems and for developing strategies to manage and mitigate their effects. Furthermore, FRP is often and more easily measured from satellites compared to CO, and determining a biome specific CO-to-FRP ratio can help to determine the daily total emissions of fires."

In response to the feedback from both reviewers, we have implemented significant revisions to the manuscript. We reviewed the manuscript, revising sections to enhance its quality by eliminating repetitions, rectifying typographical errors, and refining the wording.  Part of the changes include a compete revision of section 5.1 "CO emissions over the past two decade": We changed Figure 9 to include all inventories discussed in this paper (with the exception of GFFEPS that is currently only available for 2019). We included a figure, showing the change of emissions for different regions from all inventories discussed in this paper (GFAS, GFED, FINN 1.5, FINN2.5). Additionally, we provided a table with the trends and indicate whether or not the trend is significant (for all 5 inventories, including our own).

We provided a PDF document illustrating the differences between the previous and current versions.

To sum up, I don't see sound contributions from this study, thus I would not recommend it for publication in ACP.

---

## Referee Report (RR1)

**Review of manuscript "Towards an improved understanding of wildfire CO emissions: a satellite remote-sensing perspective" by Griffin et al.**
**Revised version.**

This manuscript describes a methodology to evaluate CO emission coefficients (EC) that may be used to estimate CO biomass burning emissions from observations of the fire radiative energy (FRE). This method relies on an inversion of CO emissions from TROPOMI satellite observations of total columns CO for a large set of fires detected in different biomes.
A first section describes the methodology for CO emissions inversion as well as an analysis of the corresponding uncertainty. The derived EC are then presented. Finally, the method is applied to the 2003-2021 time period based on the GFAS FRE database.

This study brings a valuable contribution to the efforts on improving the estimates of biomass burning emissions, for which the uncertainty is still estimated to be about a factor of 2. The approach chosen is original because it targets ECs and not just final emissions, so that it may be used in other emission inventories and time periods not covered by CO observations. I think this work is worth publishing in ACP, and that the authors did a significant effort to improve the manuscript and add new material to respond to the reviewers' comments. However, I also think that major revision is necessary before publication, which I'm confident the authors will be able to address.

**Major comments.**

I think the title should be revised to be closer to the added contribution of the manuscript, i.e. information on CO emission coefficient derived from TROPOMI observations. It is not clear if emissions are improved, which does not prevent the manuscript from making an interesting contribution.

**Emissions inversion:**
*Section 2.3, uncertainty evaluation using synthetic data:*
The authors were able to construct a database of 208 fires between May and September 2019, and only 105 remain after the filtering. It's not clear to me how these fires are representative, especially for regions in which the fire season is not during boreal spring and summer and for large fires (both very low wind and high wind may be favorable to large fires, e.g. during heatwaves).
Several criteria are listed for successful retrieval (l. 230—240 & FRP > 1000MW): are all inversions successful if these criteria are met?
*Section 2.5, evaluation top-down vs bottom up:*
In this section, the authors state that about 5000 fires are analysed (l.336); but approx. 4000 fires according to the legend of Fig.4… What is the exact number?
How many fires were detected during that time period, in total, and how many above the threshold of 1000MW (which is already very high)? What fraction of fire detection may be analysed using this method?

**EC estimates:**
This estimate relies on a landuse map to attribute specific fires to different biomes. For this purpose, they use the GLC2000 database. How is this quite old land use map representative

of vegetation in 2019-2021? I think this adds serious doubts to the results presented since it has been demonstrated that vegetation attribution is also a very large contribution to uncertainties in emission estimates (e.g. Turquety et al., GMD, 2020). There are many more recent land use classifications that may be used (e.g. from MODIS).

I think that table 2 is the most important result of the manuscript. Since the same approach is used in other work (e.g. GFAS), it would be necessary to compare results to previously published values. I understand that the authors mention that a coarser classification of biomes degrades the correlation coefficients, but a comparison is necessary in order to understand the potential added value of this work in estimating uncertainties on EC used in the literature. If I understand correctly, GFAS uses a conversion coefficient to estimate dry matter consumed from the FRE; and then derives emissions using tabulated emission factors. An equivalent emissions coefficient for CO could be estimated (or at least an order of magnitude).

**Uncertainty analysis:**
I appreciated finding an analysis of the method's uncertainties and of the uncertainties on the final emissions, which is a very difficult exercise.
It is evaluated through:
Section 2.3: academic case study with synthetic data allow an estimate of uncertainty on CO emission's inversion to 42%.
Section 3: detailed comparison of inversions with bottom-up inventory GFFEPS
Section 5: intercomparison of annual emissions for 2003-2021 using 5 other emissions inventories.
Throughout the paper, the authors estimate an uncertainty to 40% or 42% (a consistent number would probably be better). However, I think it is strongly underestimated.
This estimate assumes that the only uncertainty in the calculation of the EC values is the uncertainty on the inversion but I don't think that this is fully demonstrated. For example, the authors mention that an inversion is not possible for large wildfires because of multiple plumes. These wildfires are likely to emit a very large mass of CO. May this filter (and others) induce a bias? Does the fraction of fires with successful inversion depend on the biome?

The authors discuss the difficulty of such exercise due to overpass times and plume transport (p. 14). Would the very classic approach of comparing plumes simulated using a CTM may be more adapted in this case? It allows to compare the resulting enhancements regardless of under-constrained parameters (like diurnal variations in this case).

The authors mention that there are no uncertainty estimates for other inventories which is not true. Many publications use these inventories to simulate BB plumes using chemistry-transport models, that are compared to atmospheric observations. Some studies are included in the publications describing the inventories. It is commonly assumed that uncertainties on BB emissions are a least a factor of 2.
Different recent papers present an intercomparison of BB inventories and discuss uncertainties, e.g. :
- Wiedinmyer, C., Kimura, Y., McDonald-Buller, E. C., Emmons, L. K., Buchholz, R. R., Tang, W., Seto, K., Joseph, M. B., Barsanti, K. C., Carlton, A. G., and Yokelson, R.: The Fire Inventory from NCAR version 2.5: an updated global fire emissions model for climate and chemistry applications, Geosci. Model Dev., 16, 3873–3891, https://doi.org/10.5194/gmd-16-3873-2023, 2023.

- Pan, X., Ichoku, C., Chin, M., Bian, H., Darmenov, A., Colarco, P., Ellison, L., Kucsera, T., da Silva, A., Wang, J., Oda, T., and Cui, G.: Six global biomass burning emission datasets: intercomparison and application in one global aerosol model, Atmos. Chem. Phys., 20, 969–994, https://doi.org/10.5194/acp-20-969-2020, 2020.

Lastly, many conclusions are vague (e.g. l.338, 355). Although they are mostly supported by figures and tables I would have appreciated a few summary numbers.

**Global inventory and intercomparisons:**

Recent intercomparison exercises should be mentioned and discussed (see previous comments).

The authors find the highest agreement in trends with GFAS which may not be surprising since both approaches rely on the same FRE database. These are not independent estimates. Since the vegetation is assumed constant, the trends obtained reflects the trends in FRE.

To better understand the differences obtained for some regions in total emissions (e.g. BONA, SHAF, EQAS), it would be important to compare the values of EC used in GFAS with the values used in this study (see previous comment).

**Minor comments.**

Throughout the manuscript: check remaining typos, extra spaces in front of points, etc. I think it would be more accurate to talk about "emission inventory" rather than "emission budget"?

Abstract:
l. 4: remove 'More recently' since first emissions inversion were performed before the approaches based on FRP…
Main conclusions should provide some key numbers, as well as uncertainties.

Introduction:
l. 54, 75: laboratory measurements and field experiments.

Section 2.5 should introduce all emission inventories used in the manuscript.
What is the horizontal resolution of GFFEPS?

Section 3: 2nd sentence should be revised.

---

## Author Response (AR2)

We would like to thank Reviewer 2 for their second review. In response to the feedback from both reviewers, we have implemented significant revisions to the manuscript. Our answers are highlighted in red font, while the reviewers' comments are in black.

Here I share an anonymized and sometimes paraphrased summary of Reviewer 2's comments to the editor on the revised manuscript, which I think would be relevant in another revision round. I would be interested in author responses to these comments if a revision is pursued.

Reviewer 2 expressed overall concern that the revised manuscript failed to demonstrate the manuscript is ready for publication.

There are concerns that language in the abstract had not been revised to avoid repetitions and typos, and for sentence integrity, as requested in the first round of reviews, e.g.:

"using inputs such fuel type",

"wildfire emissions have been demonstrated to be estimated directly".

The reviewer strongly suggests more effort should be made to improve the writing and readability of the manuscript throughout.

We have made the following changes the abstract to address these concerns:

"These emissions can be estimated by a bottom-up approach that relies on fuel consumed and standardized emission factors. Emissions are also commonly derived with a top-down approach, using satellite observed fire radiative power (FRP) as proxy for fuel consumption. Wildfire emissions can also be estimated directly from satellite trace gas observations, including carbon monoxide (CO)."

Reviewer 2 also expressed that many vague statements are included instead of quantitative statements, e.g.:

From the abstract: "The CO emission totals derived from satellite data align reasonably well with those from bottom-up emission inventories for various global regions. However, notable discrepancies are evident in specific regions, such as Southern Hemisphere South America, Southern Hemisphere Africa, and Southeast Asia". According to this, the comparison is poor (but we don't know how poor) for the regions with the most fires and, thus, the most CO emissions. The reviewer therefore has concerns regarding how well the results from this study represent actual emissions.

Specifically: "...emissions [] align reasonably well with ...". "Reasonably well" is not informative; difference values, either absolute or relative, should be provided. Same for "notable discrepancies".

In the few instances where the manuscript was modified to correct this issue, the reviewer expressed concern that the new wording is often still vague (e.g., "Many of the fitted emissions" becomes "a substantial portion of the retrieved emissions", "are very close" becomes "closely matches"). The reviewer desired more effort to use quantitative statements rather than vague, subjective language.

We included the changes as suggested.

Line 2: "global CO wildfire emissions have, on the whole, decreased". Line 30: "Given the increase in fire intensity and number of fires". Readers may find these statements contradict each other but no attempt is made to address the contradiction.

The sentence about increased intensity and number of fires was talking about fires in North America. To avoid confusion we removed the sentence in question.

The reviewer shared several specific comments on using TROPOMI retrievals with qa_value <1 without applying the averaging kernels (AK) follow:

From the responses: "We also found a recent publication by Rowe et al., 2022 (https://pubs.acs.org/doi/full/10.1021/acsearthspacechem.2c00048) that investigates the uncertainty of the TROPOMI CO observations in smoke, we added the reference to our manuscript, showing approximately 10% higher CO than the aircraft measurements. This uncertainty is consisted with the uncertainty assumed in our error budget analysis (Table 1)." Rowe et al. (2022) found that TROPOMI data was higher than aircraft data by 36%. Only after correcting the TROPOMI data to account for transport and applying the AK did the TROPOMI positive bias go down to 10%.

The statement "The CO averaging kernel from the TROPOMI observations predominantly registers values close to 1 within the boundary layer" as used in the response and in the revised manuscript is incorrect. Plots in the TROPOMI L2 user manual for CO shows that the AK values within the boundary layer depart from 1 when clouds are present; several peer reviewed articles include similar plots illustrating this concept.

"the presence of clouds diminishes the sensitivity of the averaging kernel beneath them" This language could use refining. The reviewer notes that AK themselves have no sensitivity, they indicate/represent/describe sensitivity. The sentence "attempting to correct the averaging kernel in regions near fires would introduce additional uncertainty to the analysis" further suggests that the AK concept is not well understood, or the authors need to clarify what they mean. The reviewer argues that AK themselves are not to be corrected; by applying the AK to the TROPOMI retrievals it is the retrievals that are corrected. Rowe et al. (2022) findings show that correcting the TROPOMI by (among other things) applying the AK reduces the TROPOMI positive bias by 26% (please see above); it does not introduce additional uncertainty.

"Consequently, including observations with a quality flag of 1 would result in the exclusion of a substantial number of data points, primarily due to the influence of smoke." Can the authors confirm their language here? The authors say that "including data" will result in the "exclusion of a substantial number of data points". Is the wording incorrect?

"Thus, the averaging kernel is not considered in this study, but is taken into account in the overall uncertainty (uncertainty of VCDs) of the emission estimate (see Table 1). Rowe et al. (2022) investigated TROPOMI CO in thick fire plumes and found agreement within 10 % with the aircraft observations, which has been used to estimate the overall uncertainty of the emissions." The reviewer expresses concern that this statement is incorrect, since the Rowe et al. 10% agreement is based on applying the Aks.

Overall, the reviewer expresses concern that the revisions failed to demonstrate that not applying TROPOMI averaging kernels to retrievals with qa_value < 1 is the best (or at least acceptable) approach. The reviewer expressed their opinion that applying the AK in those cases is advised in the TROPOMI CO documentation and is standard practice in studies described in peer reviewed articles.

Could the authors please summarize a response/rebuttal to the reviewer's comments on TROPOMI AK usage here? Is there a misunderstanding in how the data has been assessed and processed that can be resolved?

We examined the averaging kernel and the effect of it more closely. Rowe et al (2022) used aircraft measured profiles to examine the impact of AVKs. Other than these we do not have profiles inside the fire plumes that can be applied to correct the AVK. We were looking at different averaging kernels and different profiles, see figure below, depending on the model profile that is applied to the averaging kernel the VCDs change: The stringer the enhancement near the surface the higher the VCDs are, for a plume with an enhancement of 0.4ppm near the surface the impact on the VCDs are small (0 if the the averaging kernel is 0.7 near the surface and 0.03 mol/m2 if the averaging kernel goes to 0.2 near the surface), however, if the enhancement in the applied profile is larger (shown is 1.4 ppm) the VCDs are approximately 0.1-0.14 mol/m2 higher than the original. The profiles are taken from a GEM-MACH model run (10x10 km ) with fire emissions (GFFEPS), same fire as in Figure 5 (a) which has a great overlap with the TROPOMI observed plume. Note that this GEM-MACH model output of fire plume profiles is not available globally at present.

Rowe et al. applied the averaging kernel and examined the effect using aircraft observations and aircraft observed profiles (which are not available for all fire globally). In his paper he notes that the averaging kernel improves the agreement but to a lesser effect than other (such as FLEXPART): " When the additional corrections are applied (FLEXPART, and to a lesser degree also AVK)," . Table 3 give more insight into the specific effect of the comparison between TROPOMI VCDs and aircraft observations showing that the difference without considering averaging kernels (but including other correction such as viewing geometry) leads to 13-16% difference between the aircraft and the TROPOMI CO VCDs. Based on this we increased the uncertainty due to the TROPOMI VCD to 17% which made the total uncertainty

44% instead of 42%.

[Figure]

Figure B1. Effect of applying the TROPOMI averaging kernel for a fire that overlapped well with the GEM-MACH model (using GFFEPS fire emissions). The averaging kernel corrected columns are estimated using the corrected averaging kernel AVKcorr = int(AVKz · Nz dz)/int( Nz dz) that is then applied to the CO colummns: VCDcorr = VCD/AVKcorr, where Nz is the GEM-MACH CO profile. The original CO columns are shown on the left panel, the corrected ones in the middle panel, and the difference (VCD-VCDcoor) on the right panel. The emissions were estimated with the original and corrected VCDs and show an increase of 17 % when the corrected VCDs are

used. The example shown is for the same fires as in Fig. 5 top panel (May 21, 2019 at 57◦N and 118◦ ).

Section 2.1:

"Consequently, only including observations with a quality flag of 1 would result in the exclusion of a substantial number of data points, primarily due to the influence of smoke (we found most pixels inside smoke plumes have a quality flag value of 0.7). The CO averaging kernel from the TROPOMI observations predominantly registers values close to 1 within the boundary layer for cloud-free conditions,  specifically around 0.95 with a narrow range of variability (approximately ±0.05) (Schneising et al., 2020). Nevertheless, the presence of clouds diminishes the sensitivity of the averaging kernel beneath them. It is important to note that smoke is primarily composed of  fine particles, 0.25 um or smaller. (∼1 µm).  At a wavelength of 2.3 µm these small particles scatter minimal light.   Looking at the TROPOMI averaging  kernel, we found that in case of fires the sensitivity close to the surface is typically lower than 1. Rowe et al. (2022) investigated TROPOMI CO in thick fire plumes and found agreement within 13-16 % (Table 3) without considering the averaging kernel which has been used to estimate the overall uncertainty of the emissions. The effect of the averaging kernel depends on 1) the shape of the averaging kernel and 2) on the CO profile, looking at different profiles and averaging kernels, we found the largest effect is for an averaging kernel that is close to 0 at the surface and for a strong enhancement of the CO profile, the magnitude of this enhancement determines the magnitude of the averaging kernel corrected columns. We found that the effect that applying the averaging kernel inside the smoke plume always increases the CO columns. Other than (Rowe et al., 2022) (who investigated fires during FIREX-AQ) we do not have profiles (globally) that can be applied for an averaging kernel correction. Testing the effect on the emissions we used GEM-MACH model profiles and applied the averaging kernel correction (see Appendix Fig. B1) which showed a 17 % increase of the emissions, and thus we attribute 17 % uncertainty due to the uncertainties of TROPOMI CO columns (the uncertainties are further discussed in Sect. 2.3)."

"Geostationary satellite sensors, such as TEMPO (covering North America), Geostationary Environment Monitoring Spectrometer (GEMS), or Sentinel-4 (covering Europe and Africa) will help to validate the diurnal pattern of emissions." The type of emissions should be clarified. Please note that neither TEMPO nor GEMS measure CO and CO is not among the species to be measured by Sentinel-4.
We have included the following to the sentence:
"…will help to validate the diurnal pattern of emissions (for example for NO2 and HCHO, note that none of the current geostationary satellites have the ability to measure CO)."

"This discrepancy is likely due to an underestimation from GFFEPS rather than an overestimation of TROPOMI emissions [...]" Rowe et al. (2022) showed that TROPOMI retrievals overestimate CO from fires by 36% (10% corrections, including AK correction). That could arguably result in an overestimation of TROPOMI emissions.
See averaging kernel discussion above -> applying the AK leads to higher total columns and higher emissions.

The reviewer also noted some specific comments about how emission coefficients (EC) are derived from TROPOMI and MODIS:

• "Using emission coefficients from 2019,2020, and 2019-2021 combined did not impact the total emissions (see Fig. C1), only for 2021 the total emissions reduced by approximately 20 %" Fig. C1 does not support this statement: SHAF (South. Hem. Africa), depending on what year is used as the reference (either 2019, 2020, or 2021), annual emissions vary by > 20 Mt, a 25% departure from average values. NHAF (North. Hem. Africa) vary by 20 Mt, or 40%. Global emissions vary by ~ 100 Mt, or 33%. The reviewer expressed concern therefore that contrary to what is stated in the revised manuscript, the magnitude of the emission coefficients will vary depending on what year is selected to calculate them. Also included other biomes in addition to GLC2000 , the effect is xx. The uncertainty is expected to be 40% based on the variation on the emission coefficients thus all the values previously stated are with the estimated uncertainties.

• Focusing on the emissions coefficient (EC) calculated for biome #1: Tree cover, broadleaved, evergreen. This biome is 1) one of the most extensive by area and 2) covers regions with the most CO emissions from fires, e.g., Amazon, Equatorial Africa, Indonesia, etc. Depending on the year considered, the EC calculated from TROPOMI and MODIS for biome #1 goes from 150 (2019) to 101 (2020) to 59 (2021), according to Tables C1, C2, and C3. That is an absolute change in EC=91 or a relative change in EC=88%. This shows that TROPOMI-derived EC values are highly dependent on what year is used to calculate them. The opposite is argued in the manuscript over the years, we conducted an extensive analysis of the entire time series and calculated CO emissions spanning from 2003 to 2021")
The latter comment may again be a question of using more qualitative language when quantitative language would be preferred. What do the authors mean by "relatively stable over the years?". I think the reviewer is wondering what metric is used to assess stability, given there is a reasonably large range in interannual EC.

Based on this concern we included the following discussion:

"To assess the uncertainty of the total annual emissions of our estimates  (TROPOMI-FRE), we also used emission coefficients derived from fires of individual years (2019 to 2021, using the GLC2000 classification). Using emission coefficients from 2019,2020, and 2019-2021 combined did not impact the total emissions for individual regions or globally (see Fig. E1). 2021 seems to be the anomaly for which the total global emissions reduced by approximately 20-25 % , due to overall lower EC_CO (for biomes 1-3, see Table E3 which affected the SHAF region the most and almost halved the emissions in 2021 compared to using the ECs from other years). This shows that the uncertainty of our approach is at least 25 % for the global emissions but for individual regions an uncertainty of 50 % should be assumed. The impact of using ECs from individual years is greater than the impact of using different definitions of biomes. It should be noted that the uncertainty discussed here is only due to the ECs, the uncertainty due to GFAS FRE (not provided) does not go into this estimate thus the total uncertainty of the TROPOMI/FRE top-down emissions is expected to be higher. For most wildfire emissions inventories an uncertainty of a factor of two is assumed (Pan et al., 2020; Wiedinmyer et al., 2023)."

We would like to thank Reviewer 3 for their review. In response to the feedback from both reviewers, we have implemented significant revisions to the manuscript.
Our answers are highlighted in red font, while the reviewers' comments are in black.

**Review of manuscript "Towards an improved understanding of wildfire CO emissions: a satellite remote-sensing perspective" by Griffin et al.**
**Revised version.**
This manuscript describes a methodology to evaluate CO emission coefficients (EC) that may be used to estimate CO biomass burning emissions from observations of the fire radiative energy (FRE). This method relies on an inversion of CO emissions from TROPOMI satellite observations of total columns CO for a large set of fires detected in different biomes.
A first section describes the methodology for CO emissions inversion as well as an analysis of the corresponding uncertainty. The derived EC are then presented. Finally, the method is applied to the 2003-2021 time period based on the GFAS FRE database.
This study brings a valuable contribution to the efforts on improving the estimates of biomass burning emissions, for which the uncertainty is still estimated to be about a factor of 2. The approach chosen is original because it targets ECs and not just final emissions, so that it may be used in other emission inventories and time periods not covered by CO observations. I think this work is worth publishing in ACP, and that the authors did a significant effort to improve the manuscript and add new material to respond to the reviewers' comments. However, I also think that major revision is necessary before publication, which I'm confident the authors will be able to address.

**Major comments.**
I think the title should be revised to be closer to the added contribution of the manuscript, i.e. information on CO emission coefficient derived from TROPOMI observations. It is not clear if emissions are improved, which does not prevent the manuscript from making an interesting contribution.
The title has been changed to reflect this suggestion: "Forest Fire CO Emissions: Exploring Insights through TROPOMI-derived Emissions and Emission Coefficients"
**Emissions inversion:**
*Section 2.3, uncertainty evaluation using synthetic data:*
The authors were able to construct a database of 208 fires between May and September 2019, and only 105 remain after the filtering. It's not clear to me how these fires are representative, especially for regions in which the fire season is not during boreal spring and summer and for large fires (both very low wind and high wind may be favorable to large fires, e.g. during heatwaves).
Several criteria are listed for successful retrieval (l. 230—240 & FRP > 1000MW): are all inversions successful if these criteria are met?
All of section 2.3 is based on synthetic columns to test the accuracy of the direct emission estimate. This is done to determine the uncertainty of the method. The model is available at a resolution of 10x10km only in North America that is needed to be comparable to the satellite resolution. These are all synthetic and do not go into any estimates in the later sections. The sole purpose is to estimate the uncertainty of the method. For the emissions we apply the direct emission estimate and then apply the filter. Specific number and more details have been included in the manuscript in Sect. 2.4:

"For a typical year using the described MODIS clustering, we are left with approximately 13-18 thousand fire clusters globally for which we attempt an emission estimate. For about 3-4 thousand fires the estimate fails entirely. And further 9-12 thousand fire emissions are filtered due to poor quality, leaving 4-6 thousand successful fire emissions globally per year."

*Section 2.5, evaluation top-down vs bottom up:*

In this section, the authors state that about 5000 fires are analysed (l.336); but approx. 4000 fires according to the legend of Fig.4… What is the exact number?

How many fires were detected during that time period, in total, and how many above the threshold of 1000MW (which is already very high)? What fraction of fire detection may be analysed using this method?

1000MW (1GW) is the threshold using the sum of all hotspots of the fire cluster and it's not a very large number; to make this clearer we included the term "fire cluster" in the manuscript. As can be seen in Fig.6 fires with FRP<1GW (=1000MW) are not really relevant for the estimate of the emission coefficient (scale is from 1 to 100GW). I would also glice to point out that the minimum number of hotspots is set to 5. Reducing the minimum FRP threshold or number of hotspots will lead to more failed fires as those are likely the TROPOMI detection limit, or false positives such as flaring from industry). We included the following in the manuscript to make this clearer:

Sect. 2.4:

…"1 GW (note, this threshold is not for individual hotspots but for the fire cluster) and a confidence of at least 75 % (for individual hotspots).

**"EC estimates:**

This estimate relies on a landuse map to attribute specific fires to different biomes. For this purpose, they use the GLC2000 database. How is this quite old land use map representative

of vegetation in 2019-2021? I think this adds serious doubts to the results presented since it has been demonstrated that vegetation attribution is also a very large contribution to uncertainties in emission estimates (e.g. Turquety et al., GMD, 2020). There are many more recent land use classifications that may be used (e.g. from MODIS). : *The reviewer is correct in stating there are other, more recent land use classifications than GLC2000. The decision to use GLC2000 was to remain consistent with that used by GFEPPS . We needed a global land use of sufficient resolution that was easy to employ (for proof of concept) and GLC2000 was well suited. It provided single map global coverage at a 1-km resolution. We considered the reviewers concerns and added the emission coefficients using MODIS land classification ( 0.05 degree resolution), as well as GFED land classification (0.25 degree resolution, and only 6 types of vegetation).*

*In terms of timeliness, this was seen as less critical as vegetation changes relatively slowly and most land use changes, whether they were a result of disturbance (fires, deforestation) or urbanization, would result in landscapes less fire prone and would be reflected by a reduced number of hotspots.*

*We modified section 5 (Figure 8, and text) to include the estimated emissions using the emission factors from different biome classifications (GLC2000, MODIS, GFED).*

*"Section 4:"* To differentiate biomass burning emissions in different biomes we use the GLC2000 (European Commission, 2003) as used for the development of GFFEPS, additionally we also tested the classification from MODIS (MCD12C1) and the GFED 425 partitioning.

[...] The results using the MODIS MCD12C1 land classification instead are shown in Table 3. For water, snow/ice, urban, and sparsely vegetated no EC has been derived as there were unsurprisingly no fires found in the direct emissions from TROPOMI on these land types. As expected the EC for broadleaf evergreen forest are the highest (as for the GLC2000 definition) with 129g/MJ and the lowest are found for closed shrublands (23 g/MJ).

GFED relies on only 6 different types of vegetation (AGRI: Agricultural waste burning; BORF: Boreal forest fires; DEFO: Tropical deforestation and degradation; PEAT: Peat fires; SAVA:Savanna, grassland, and shrubland fires; and TEMF: Temperate forest fires) and the results of the emission coefficients using the TROPOMI direct emissions estimates (and MODIS FRP) are shown in Table 4. This gives the opportunity to be able to compare with the emission factors from GFED and GFAS directly  (shown in Table F1 although the conversion factor to convert the ECs to Efs can change by a factor of four depending on the study used (Wooster et al., 2005; Kaiser et al., 2012), limiting a meaningful comparison). From our analysis: PEAT the highest Ecs (183 g/MJ), and the lowest are found for agricultural waste burning."

*Section 5:"* The total wildfire related CO emissions using TROPOMI-FRE are approximately 290 Mt in 2019. Using different land classifications we get 308 Mt using the MCD12C1 classification and 265 Mt for GFED classification, these help to determine an overall uncertainty for the methodology and the total emissions of the TROPOMI-MODIS top-down emissions. Using the coarser classification of GFED leads to lower emissions (~ 15Mt) in the SHSA and SHAF region compared to using the ECs for GLC2000 or MCD12C1 classification."

I think that table 2 is the most important result of the manuscript. Since the same approach is used in other work (e.g. GFAS), it would be necessary to compare results to previously published values. I understand that the authors mention that a coarser classification of biomes degrades the correlation coefficients, but a comparison is necessary in order to understand the potential added value of this work in estimating uncertainties on EC used in the literature. If I understand correctly, GFAS uses a conversion coefficient to estimate dry matter consumed from the FRE; and then derives emissions using tabulated emission factors. An equivalent emissions coefficient for CO could be estimated (or at least an order of magnitude).

We included emission coefficients using the GFED biome classification (Table 4), additionally we converted these to emission factors to compare to GFED and GFAS emission factors directly (in appendix Table D1). Note that the conversion factor to estimate emission factors varies significantly for different studies, by as much as a factor of 4. We added the following to the manuscript:
"Section 4: The results using the MODIS MCD12C1 land classification instead are shown in Table 3. For water, snow/ice, urban, and sparsely vegetated no EC has been derived as there were unsurprisingly no fires found in the direct emissions from TROPOMI on these land types. As expected the EC for broadleaf evergreen forest are the highest (as for the GLC2000 definition) with 129g/MJ and the lowest are found for closed shrublands (23 g/MJ). 460 GFED relies on only 6 different types of vegetation (AGRI: Agricultural waste burning; BORF: Boreal forest fires; DEFO: Tropical deforestation and degradation; PEAT: Peat fires; SAVA:Savanna, grassland, and shrubland fires; and TEMF: Temperate forest fires) and the results of the emission coefficients using the TROPOMI direct emissions estimates (and MODIS FRP) are shown in Table 4. This gives the opportunity to be able to compare with the emission factors from GFED and GFAS directly (shown in Table F1 although the conversion factor to convert the Ecs to Efs can change by a factor of four depending on the study used (Wooster et al., 2005; Kaiser et al., 2012), limiting a meaningful comparison). From our analysis: PEAT the highest Ecs (183 g/MJ), and the lowest are found for agricultural waste burning."
Section 5:" The total wildfire related CO emissions using TROPOMI-FRE are approximately 290 Mt in 2019. Using different land classifications we get 308 Mt using the MCD12C1 classification and 265 Mt for GFED classification, these help to determine an overall uncertainty for the methodology and the total emissions of the TROPOMI-MODIS top-down emissions. Using the more coarse classification of GFED leads to lower emissions ($\sim$ 15Mt) in the SHSA and SHAF region compared to using the Ecs for GLC2000 or MCD12C1 classification."

**Uncertainty analysis:**
I appreciated finding an analysis of the method's uncertainties and of the uncertainties on the final emissions, which is a very difficult exercise.
It is evaluated through:
Section 2.3: academic case study with synthetic data allow an estimate of uncertainty on CO emission's inversion to 42%.
Section 3: detailed comparison of inversions with bottom-up inventory GFFEPS
Section 5: intercomparison of annual emissions for 2003-2021 using 5 other emissions inventories.
Throughout the paper, the authors estimate an uncertainty to 40% or 42% (a consistent number would probably be better).

However, I think it is strongly underestimated.

This estimate assumes that the only uncertainty in the calculation of the EC values is the uncertainty on the inversion but I don't think that this is fully demonstrated. For example, the authors mention that an inversion is not possible for large wildfires because of multiple plumes. These wildfires are likely to emit a very large mass of CO. May this filter (and others) induce a bias? Does the fraction of fires with successful inversion depend on the biome?

We made sure the number is consistent. We added the following to Sect. 5:

"To assess the uncertainty of the total annual emissions of our estimates (TROPOMI-FRE), we also used emission coefficients derived from fires of individual years (2019 to 2021, using the GLC2000 classification). Using emission coefficients from 2019,2020, and 2019-2021 combined did not impact the total emissions for individual regions or globally (see Fig. E1). 2021 seems to be the anomaly for which the total global emissions reduced by approximately 20-25 % , due to overall lower EC_CO (for biomes 1-3, see Table E3 which affected the SHAF region the most and almost halved the emissions in 2021 compared to using the Ecs from other years). This shows that the uncertainty of our approach is at least 25 % for the global emissions but for individual regions an uncertainty of 50 % should be assumed. The impact of using Ecs from individual years is greater than the impact of using different definitions of biomes. It should be noted that the uncertainty discussed here is only due to the Ecs, the uncertainty due to GFAS FRE (not provided) does not go into this estimate thus the total uncertainty of the TROPOMI/FRE top-down emissions is expected to be higher. For most wildfire emissions inventories an uncertainty of a factor of two is assumed (Pan et al., 2020; Wiedinmyer et al., 2023)."

Yes, the success-rate of the direct emission estimate does depend on the biome. For example, the large fires during the black summer in Australia would largely affect biome type 1. However when the EC are estimated (sect. 4), only successful fires with total FRP of those specific fires are considered. No inversion is used for the annual total emissions. Fires that were not successfully estimated from TROPOMI, but detected by MODIS are part of GFAS FRE (here, no minimum FRP threshold is applied) and are still considered in the total emissions (sect. 5) through the use of the GFAS FRP and applying the emission coefficients that are derived in section 4. We are not applying an inversion technique in this paper to quantify total fire emissions, rather we are using 1) direct emission estimates (flux method) based on winds and observed total column CO, and 2) top-down estimates by applying TROPOMI/MODIS-derived EC to total FRP. Throughout the text we refer to "direct" estimates and "top-down" estimates to distinguish the two.

We included changes in the abstract, Section 3, Section and Conclusions to make this clearer.

The authors discuss the difficulty of such exercise due to overpass times and plume transport (p. 14). Would the very classic approach of comparing plumes simulated using a CTM may be more adapted in this case? It allows to compare the resulting enhancements regardless of under-constrained parameters (like diurnal variations in this case).

We have used synthetic total column CO from a CTM in sect. 2.3 to obtain better understanding of the uncertainties associated with the direct emission estimate and to explore how best to filter the emission estimates. Figure 2 shows how the estimate (using the model CO columns) compares with the input emissions to the CTM. Which allows to evaluate the emission estimate method directly.

We also compare some plumes, from the model with TROPOMI (total columns), see Fig.5, however this kind of comparison might be difficult to do globally (we have no global run GEM-MACH version available at 10km resolution), and often the accuracy of the model meteorology plays an important role in how similar the plumes in reality vs model look like.

The diurnal variability is included in the model emissions, but TROPOMI alone cannot provide the diurnal variability.

The authors mention that there are no uncertainty estimates for other inventories which is not true. Many publications use these inventories to simulate BB plumes using chemistry-transport models, that are compared to atmospheric observations. Some studies are included in the publications describing the inventories. It is commonly assumed that uncertainties on BB emissions are a least a factor of 2.

Different recent papers present an intercomparison of BB inventories and discuss uncertainties, e.g. :

Wiedinmyer, C., Kimura, Y., McDonald-Buller, E. C., Emmons, L. K., Buchholz, R. R., Tang, W., Seto, K., Joseph, M. B., Barsanti, K. C., Carlton, A. G., and Yokelson, R.: The Fire Inventory from NCAR version 2.5: an updated global fire emissions model for climate and chemistry applications, Geosci. Model Dev., 16, 3873–3891, https://doi.org/10.5194/gmd-16-3873-2023, 2023. Pan, X., Ichoku, C., Chin, M., Bian, H., Darmenov, A., Colarco, P., Ellison, L., Kucsera, T., da Silva, A., Wang, J., Oda, T., and Cui, G.: Six global biomass burning emission datasets: intercomparison and application in one global aerosol model, Atmos. Chem. Phys., 20, 969–994, https://doi.org/10.5194/acp-20-969-2020, 2020.

We included the suggested publications and added the following sentence in Section 5:

"For most wildfire emissions inventories an uncertainty of a factor of two is assumed (Pan et al., 2020; Wiedinmyer et al., 2023)."

Lastly, many conclusions are vague (e.g. l.338, 355). Although they are mostly supported by figures and tables I would have appreciated a few summary numbers.

We have made revision to the conclusions as suggested:

"The comparison of TROPOMI-FRP derived top-down emissions with other inventories reveals disparities and highlights the substantial uncertainties associated with fire emission estimates. Notably, GFFEPS generally exhibits the closest agreement with TROPOMI-FRE (e.g. total emissions for 2019: 337 vs 290Mt of CO for GFFEPS versus TROPOMI-FRE, respectively) , although some exceptions are evident in regions such as SHSA, NHSA, SEAS, and AUST. Possible reasons for the discrepancy are the omission of small fires, the absence of detected hotspots, and an underestimation of FRP, ultimately contributing  to an overall underestimation of total emissions. FINNv2.5 shows the largest discrepancies with TROPOMI-FRE where the FINNv2.5 global emissions (579 Mt) are almost twice of TROPOMI-FRE, in the SEAS region the FINNv2.5 emissions are higher by a factor of five: 20Mt versus 100 Mt). Examining the trends over the past two decades (corresponding to the MODIS lifetime), it appears that global CO wildfire emissions have, on the whole, decreased, the trend (between -8.7 to -5.1 Mt/yr) is significant for all inventories, except GFED (this is driven by the high increase for BOAS of almost 20Mt/yr). The trend of wildfire emissions is highly region-specific, with the highest reductions occurring in SHSA, SHAF, NHAF, and CEAS. Conversely, wildfire emissions in TENA are on the rise (by 0-4 Mt/yr). For all other regions, the variability within the past two decades has been too substantial to determine a statistically significant trend."

**Global inventory and intercomparisons:**

Recent intercomparison exercises should be mentioned and discussed (see previous comments). The authors find the highest agreement in trends with GFAS which may not be surprising since both approaches rely on the same FRE database. These are not independent estimates. Since the vegetation is assumed constant, the trends obtained reflects the trends in FRE.

To better understand the differences obtained for some regions in total emissions (e.g. BONA, SHAF, EQAS), it would be important to compare the values of EC used in GFAS with the values used in this study (see previous comment).

We included this suggestion, see previous comments.

**Minor comments.**

Throughout the manuscript: check remaining typos, extra spaces in front of points, etc. I think it would be more accurate to talk about "emission inventory" rather than "emission budget"?

We have changed emission "budget" to emission "inventory" or simply "total emissions" throughout the manuscript as suggested.

Abstract:

l. 4: remove 'More recently' since first emissions inversion were performed before the approaches based on FRP…

Removed as suggested.

Main conclusions should provide some key numbers, as well as uncertainties.

Changes to the conclusions were made see above comment. The following is in the conclusion section:

"The sensitivity tests show that the methods uncertainty is approximately 34% (and 44% total uncertainty including the uncertainty of VCDs and winds)."

"The TROPOMI-FRE top-down emissions (based on the uncertainties of the ECs) are at least 50% regional or 25% globally. Note that the total uncertainty is likely larger as the uncertainty of GFAS FRE is not accounted for."

Introduction:

l. 54, 75: laboratory measurements and field experiments.

Unfortunately, I'm not sure what the refers to.

Section 2.5 should introduce all emission inventories used in the manuscript.

We changed the name of the section to "Emission inventories" and included the following paragraph:

"We compare our retrieved CO emissions to several existing biomass burning CO inventories, namely GFAS, GFED, FINN v1.5 and v2.5, as well as GFFEPS. GFED (van der Werf et al., 2017) and FINN are both based on bottom-up approach. Here, we use GFEDv4.1 which has a 0.25◦ resolution, developed by NASA, in Sect. 5 we use the annual total emissions for different geographical regions. The FINN inventory (Wiedinmyer et al., 2006, 2011) by NCAR is not on a regular grid but based on the location of the MODIS or VIIRS detected hot spots, these are then summed to obtain annual totals in Sect.5. The GFAS fire emission inventory (Kaiser et al., 2012) by ECMWF utilizes a top-down approach based on MODIS FRP and is on a 0.1◦ regular grid, here we use v1.2. Additionally, we also compare to a new global biomass burning algorithm, GFFEPS, developed by Environment and Climate Change Canada. "

What is the horizontal resolution of GFFEPS?

The emissions are directly based on the hotspot measurements, they are not gridded. We added the following to the manuscript:

"The resulting emissions are not gridded, but distributed to the location of the detected fire-hotspots. "

Section 3: 2nd sentence should be revised.

We changed the sentence to the following:

"Stockwell et al. (2022) have shown good agreement between the TROPOMI-derived to aircraft-derived CO fire emissions as part of the FIREX-AQ campaign (Warneke et al., 2023). The sensitivity tests (Sect. 2.3) using synthetic total columns also suggest that emissions can be reliably estimated using the flux method within 42 % uncertainty.

---

## Author Response (AR3)

We would like to thank the reviewer for her/his review. We addressed each comment below, our answers and changes to the manuscript are highlighted in red. Additionally we added a reference to Anderson et al. (2024) that describes the GFFEPS emission prediction system (it was not available previously).

GENERAL COMMENTS

The manuscript makes many references to 'wildfires' and 'forest fires' (in the title) whereas the method does not select any fire types but covers basically everything. In this case I would think that 'biomass burning' would be more appropriate since it also covers controlled fires, and savannah fires.

Thanks for this suggestion, we agree that biomass burning is a better term. We have made the changes (from wildfire -> biomass burning and forest fires -> biomass burning, where appropriate) throughout the manuscript as well as in the title.

TROPOMI XCO is mentioned is a good source of independent information to test fire inventories, but the study does not actually perform such a test. Without this we still do not know which of the inventories in Figure 8 performs best. FRP is used to extrapolate TROPOMI information, but any of the other approaches could have been used for this also. It is an implicit choice that has been made without motivation. Better would have been to first test all the inventories against TROPOMI at the collected set of locations and then select the method that is most consistent with TROPOMI as a method for extrapolation.

We are a little unsure exactly what the reviewer is referring to here. But we assume they are suggesting that we use all the various inventories mentioned in our paper in the forecast model, and then compare the resultant CO with TROPOMI. This is an interesting study, but it by itself would be a very large effort and it is not the study we are engaging in. Using the output of such a study - say the inventory that produced the output that agreed best with TROPOMI - to scale up our emissions to regional and annual totals is problematic since these inventories are gridded, sometimes by hour and sometimes by day, and on varying spatial scale. this makes it very difficult to compare, and hence scale up, our emissions which are snap-shots for individual fires. Our method of using FRP to scale up or extrapolate is better suited since TROPOMI emissions and MODIS FRP can be directly linked in space and time.

In some places errors are assumed to cancel out in the mean, without proper justification. One example is the distinction between flaming and smoldering fires. Since the TROPOMI scenes that are used are quite a narrow selection of conditions for which the approach works best, it is unlikely to be representative of all fires. For example, the smoke generation in incomplete combustion is likely to be filtered out. Sampling biases are also likely to occur just because the size of fire and the meteorological conditions that are selected. I do not require to account for all this, but to spend a few sentences in the discussion to make the reader aware of such shortcomings in the current version.

As suggested, we included the following to the conclusion section:
" There is much uncertainty in this method as a single EC is assumed for each biome that is based on emission estimates at around 1:30pm local time and thus the TROPOMI scenes in general represent a small selection of conditions, it is unlikely to be representative of all fires or fire stages. The EC can also be affected by sampling biases because the size of fires and the meteorological conditions that are selected."

Another example is the error due to neglecting the vertical sensitivity of TROPOMI. No correction is made, whereas the error is always positive and therefore a bias. For this reason, it is insufficient to account for it as an uncertainty. I accept that a proper solution would require vertical profile information which can be solved later, but a 17% upward correction would be better than no correction.

Depending on the height of the plume and the averaging kernel, the effect of applying the AVK can be positive or negative. In our responses to reviewer 2 where an enhancement was where the AVK was low (near the surface) the VCDs increased, Rowe et al. (2022) found that on average the VCDs decreased when applying the aircraft observed profile. So depending on the shape of the profile and

averaging kernel the effect can increase or decrease the VCDs. Thus we are not applying a factor to correct the TROPOMI VCDs. We agree that the error is of systematic nature rather than random and added it to the random uncertainty rather than adding it in quadrature, this increased the total uncertainty to 57%.
We added the following to the text as well as changed Table 1.
 Section 2.1: " Validation against TCCON measurements around the world showed that the TROPOMI CO columns have a high bias of about 10% (Sha et al., 2021) ."
Section 2.3: " The total uncertainty of the satellite-derived emissions (see Table 1), is based on the systematic bias and random uncertainty. The random uncertainties consist of the wind speed (~ 10%), the effect of the altitude used for the wind speed (~20%), and the uncertainty of the method itself (based on the relative difference between the true and fitted emissions of 34% after applying the above mentioned quality filters).  The uncertainty of the wind speed caused by the uncertain altitude of the plume is based on the mean difference of the wind speed when comparing the winds 50\,hPa above and below the aerosol layer height.  The uncertainty of the wind speed is based on \citet{gualtieri2022} who found approximately 0.5m/s for the 90% confidence interval for ERA5, with the average wind speed of approximately 5m/s (for our dataset), we assume a 10% uncertainty for the wind speed.  These errors are added in quadrature, leading to a total random uncertainty of 41%. Additionally, the TROPOMI CO VCDs (comparison to TCCON) are biased high by about 10% (Sha et al., 2021), and not accounting for the averaging kernel correction due to the lack of profile observations for the fires will add another 6% (Rowe et al, 2022) (Table 3, difference between accounting and neglecting the averaging kernel). While the total emissions (or VCDs) could be scaled by the systematic bias, the effect of the averaging kernel correction depends on the profile as well as averaging kernel shape, also see averaging kernel analysis in Sect. 2.1 and Fig.B1). Adding the systematic and random error leads to a total uncertainty of 57%."

SPECIFIC COMMENTS

Line 73: The TROPOMI resolution is 7x5.5 km2 as mentioned later
Corrected.
Figure 1: Why use Gaussian functions rather than the observed enhancement over the background? Do not see the added value. I wonder also if the width of the integration box in panel d is wide enough for the tail of the plume. Tail dispersion outside the box could explain part of the apparent emission reduction with distance.
The Gaussian function is not used for the background correction. The Gaussian function is used to find the plume width, the correction for the wind direction, and correction of any offset from the centre (in case the estimated location from weighted MODIS thermal anomalies is not quite correct).
This is explained in the manuscript l. 151-170.
We have included the following sentence t make this clearer this in the manuscript:
" New improvements with regards to the plume rotation and plume widths are included, and illustrated in Fig. 1 where Gaussians are fitted across the plume to be able to automate the estimation by determining the plume width, correct the wind  direction and correct the centre location of the fire."
Fig. 1 caption: " (b) simple Gaussians are fitted across wind in 4-km wide boxes up to 40 km downwind of the fire to find the plume width, correct the wind direction and fire location"

Line 167: An upwind mixing ratio as background can be quite inaccurate if the plume is superimposed on a non-uniform background. This might also cause apparent temporal variations in the emissions.
We tested various methods to determine the background CO concentrations including Gaussian fit, using the 90[th] percentile and the upwind CO concentrations. These were also tested with the CTM simulation (see Section 2.3) and we found that using the average upwind concentrations was the most stable method with the best results. If the background is too large the results are filtered.

Line 257: The error calculation missing 10% VCD uncertainty from the TROPOMI retrieval. (the 17% is only from the neglected Ak)

We considered scaling our result to account for this systematic effect but in the end decided it was best for the data user to apply such a correction, either the generic 6% value from Rowe et al. (2022) or 10% from Sha et al. (2021) or perhaps one better suited to their specific application.

In Rowe et al. 2022 the difference between applying the AVK (quality assured and using AVK: 10+-15%) and not applying the AVK (quality assured 16+-15%) is approximately 6% see (Table 3), in the 16% included in this study the 10% bias from the TCCON study is included and rounded up. However, instead of assuming this is a random error we included it as a systematic error and added it to the random uncertainty (rather than adding it in quadrature). This increases the total uncertainty to 57%.

We added the following to the manuscript and increased the total error (from 44 to 57%) throughout the manuscript:

Line 348: "red dot" io "orange dot"?
Corrected.

Line 357: 'this bias' is which bias?
Changed to: " The cause of the differences between the TROPOMI-derived and GFFEPS emissions are being investigated"

Line 369: What could be learned from AUST having a very poor correlation?
The model does currently not capture the emissions from Australian eucalyptus well.
We added the following: " Other regions, like AUST have a very poor correlation, slope and RMSE, indicating a need to improve the modelling of that region, which is currently still under development, such as improving the emission factors, correcting the fuel consumption and combustion completeness for eucalyptus (Anderson et al., 2024)."

Line 416: Increases instead of reduced this error.
Reduces is the correct term here. The smoldering CO emissions are very high, but the FRP for smoldering is very low thus the increase is not as significant. We changed the term to " has a smaller effect on the total emissions" to avoid confusion.

Figure 6: "blue being the outliers" You mean purple? Not clear what 'density' means here (I suppose frequency?). What is 'S'? The plot does not look like R=0.7. The axis are plotted such that we are essentially only looking at outliers. This should be improved.
The word outlier is not used correctly, this has been corrected.
The caption has been corrected to: "The black line indicates the slope ("s") of the best fit. … (yellow being the most frequent and purple being single points)"

Table 2: With rank you mean the fraction of total RFP?
Thanks for pointing this out, we clarified this more:
We added the following to the caption in Tables 2, 3 and 4: " rank of importance in terms of total FRP with 1 being the biome contributing the most to the emissions (``Rank'')"
In the text: " rank is number of importance with respect to the total annual FRP from GFAS with 1 being the biome contributing the most (for the 2019 base year)."

Line 566: 'by this' is by what?
Somehow this sentence ended up in the wrong location, it has been moved a few lines down and corrected:
" We also found that the FRP is strongly influenced by thick smoke which can influence these types of top-down emission estimates, and leads to an underestimate of fire emissions for fires with thick smoke (typically large fires). The directly-derived TROPOMI CO emission estimates are not impacted as much by the by the smoke and have been used here to verify and analyse this issue for individual fires."

Line 586: 'are at least' what?
Corrected to: " The TROPOMI-FRE top-down emissions uncertainties (based on the uncertainties of the ECs) are at least 50% for regional emissions or 25% for global emissions."

TECHNICAL CORRECTIONS

Line 181: The sentence starting with 'Any comparisons to emissions…' has been duplicated
Corrected
Line 345: GFFEPS io GFEEPS
Corrected.
Line 370: Table D2)
Corrected
Line 444: 'evergreen'
Corrected